# Structure-based electron-confurcation mechanism of the Ldh-EtfAB complex

Kanwal Kayastha[1†‡], Alexander Katsyv[2†], Christina Himmrich[2], Sonja Welsch[3], Jan M Schuller[4], Ulrich Ermler[1*], Volker Müller[2*]

[1]Departments of Molecular Membrane Biology of the Max-Planck-Institut for Biophysics, Frankfurt am Main, Germany; [2]Department of Molecular Microbiology & Bioenergetics, Institute of Molecular Biosciences, Goethe University, Frankfurt am Main, Germany; [3]Central Electron Microscopy Facility, Max Planck Institute of Biophysics, Frankfurt am Main, Germany; [4]SYNMICRO Research Center and Department of Chemistry, Philipps University, Marburg, Germany

**\*For correspondence:**
ulrich.ermler@biophys.mpg.de
(UE);
vmueller@bio.uni-frankfurt.de
(VM)

[†]These authors contributed equally to this work

**Present address:** [‡]Biophysical Structural Chemistry, Leiden Institute of Chemistry, Gorlaeus Laboratories, Leiden University, Leiden, Netherlands

**Competing interest:** The authors declare that no competing interests exist.

**Abstract** Lactate oxidation with $NAD^+$ as electron acceptor is a highly endergonic reaction. Some anaerobic bacteria overcome the energetic hurdle by flavin-based electron bifurcation/confurcation (FBEB/FBEC) using a lactate dehydrogenase (Ldh) in concert with the electron-transferring proteins EtfA and EtfB. The electron cryo-microscopically characterized (Ldh-EtfAB)$_2$ complex of *Acetobacterium woodii* at 2.43 Å resolution consists of a mobile EtfAB shuttle domain located between the rigid central Ldh and the peripheral EtfAB base units. The FADs of Ldh and the EtfAB shuttle domain contact each other thereby forming the D (dehydrogenation-connected) state. The intermediary Glu37 and Glu139 may harmonize the redox potentials between the FADs and the pyruvate/lactate pair crucial for FBEC. By integrating Alphafold2 calculations a plausible novel B (bifurcation-connected) state was obtained allowing electron transfer between the EtfAB base and shuttle FADs. Kinetic analysis of enzyme variants suggests a correlation between $NAD^+$ binding site and D-to-B-state transition implicating a 75° rotation of the EtfAB shuttle domain. The FBEC inactivity when truncating the ferredoxin domain of EtfA substantiates its role as redox relay. Lactate oxidation in Ldh is assisted by the catalytic base His423 and a metal center. On this basis, a comprehensive catalytic mechanism of the FBEC process was proposed.

## Editor's evaluation

This paper describes a high-resolution cryoEM structure of the lactate dehydrogenase-electron transferring flavoprotein complex that provides critical insight on a basic metabolic pathway of anaerobic fermentation. What is novel is that this is the first structure of a complex wherein the dehydrogenase runs as such, rather than as a reductase. The lactate produced by glycolysis is oxidized to pyruvate with concomitant reduction of $NAD^+$ to NADH. Because electrons are being supplied by the dehydrogenase, the ETF executes confurcation in contrast to all of those elucidated so far, which function in the opposite direction to effect bifurcation.

## Introduction

Anaerobic energy metabolisms frequently operate at the thermodynamic limit of life. For energy yield optimization, various microorganisms developed early in evolution an energy-coupling process termed flavin-based electron bifurcation/confurcation (FBEB/FBEC) embedded into a modular, mostly soluble enzyme complex (*Buckel and Thauer, 2013*; *Herrmann et al., 2008*; *Li et al., 2008*; *Müller et al., 2018*). In the more familiar FBEB direction, an endergonic reduction, normally of oxidized

ferredoxin (Fd$_{ox}$), is driven by a simultaneous exergonic reduction (*Figure 1A*). This two-reaction process can also proceed in the opposite direction thereby catalyzing a confurcating event. One example is the reversible bifurcating hydrogenase (*Katsyv et al., 2021*; *Schuchmann and Müller, 2012*; *Schut and Adams, 2009*; *Wang et al., 2013*) that oxidizes hydrogen gas to two protons and two single split electrons; the latter are transferred to the two-electron acceptors, NAD(P)$^+$ and Fd$_{ox}$. In the confurcation mode H$_2$ is produced from reduced ferredoxin (Fd$_{red}$) and NAD(P)H (*Katsyv et al., 2021*; *Schuchmann and Müller, 2012*; *Wang et al., 2013*). As key player for energy coupling in FBEB/FBEC serves a flavin endowed with an extremely inverted one-electron redox potential implicating a very short-living FAD$\bullet^-$ state (*Duan et al., 2021*; *Lubner et al., 2017*). In the confurcation mode, the rate-determining endergonic reduction of FAD to FAD$\bullet^-$ (< –700 mV) by Fd$_{red}$ (E ≈ –500 mV) is strongly coupled with the exergonic reduction of FAD$\bullet^-$ to FADH$^-$ (E >+200 mV) by an electron originating from the high-potential (weak) electron donor (E≤0 mV) (*Figure 1B*; *Baymann et al., 2018*; *Nitschke and Russell, 2012*). One-electron redox potentials of E(FAD/FAD$\bullet^-$) = –911 mV and E(FAD$\bullet^-$/FADH$^-$) = +359 mV are experimentally determined for the bifurcating NADPH dependent Fd$_{ox}$:NAD$^+$ oxidoreductase (*Lubner et al., 2017*). Strongly inverted one-electron redox potentials are also reported for quinones in the related quinone-based electron bifurcation process embedded into the *bc$_1$* complex (*Crofts et al., 2013*). The FBEB mechanism is outlined in various comprehensive reviews (*Buckel and Thauer, 2013*; *Müller et al., 2018*; *Peters et al., 2016*).

The electron-confurcating lactate dehydrogenase/electron-transferring flavoprotein (Ldh-EtfAB) complex, the research subject of this report, catalyzes the endergonic oxidation of lactate to the central cellular metabolite pyruvate (*Bock et al., 1994*; *Dönig and Müller, 2018*; *Kandler, 1983*) with NAD$^+$ as electron acceptor driven by simultaneous co-oxidation of Fd$_{red}$ (*Figure 1B*; *Weghoff et al., 2015*). This FBEC reaction occurs in anaerobic sulfate-reducing and acetogenic bacteria (*Vita et al., 2015*; *Weghoff et al., 2015*). Lactate is accumulated in huge amounts as intermediate during anaerobic glucose degradation *via* glycolysis to the endproduct acetate (*Müller, 2008*).

The confurcating Ldh-EtfAB complex belongs to the bEtf family, the largest and most diverse group of FBEB/FBEC enzymes (*Buckel and Thauer, 2018a*). bEtf family members are built up of a variable dehydrogenase/oxidoreductase core and several peripherally attached EtfAB modules (*Figure 1—figure supplement 1*; *Demmer et al., 2018*; *Demmer et al., 2017*; *Feng et al., 2021*). The EtfAB heterodimer is built up of an EtfAB base composed of the N-terminal segments of EtfA and EtfB (termed domains I and III, respectively) and a module termed domain II or EtfAB shuttle domain (due to its shuttling role between two sites far away from each other) formed by the tightly associated C-terminal segment of EtfA and the C-terminal arm of EtfB (*Chowdhury et al., 2014*). Related structures were reported for non-bifurcating EtfAB and EtfAB-oxidoreductase complexes (*Leys et al., 2003*; *Roberts et al., 1996*; *Toogood et al., 2007*; *Vogt et al., 2019*). Both the rigid oxidoreductase core and EtfAB base platforms and the mobile EtfAB shuttle domain carry one FAD termed d-FAD, b-FAD and a-FAD. Their electrochemical potentials are unknown for Ldh-EtfAB but measured for the related Bcd-EtfAB complex except for the one-electron redox potential of b-FAD (*Sucharitakul et al., 2020*).

In the bEtf family, a-FAD transfers electrons between the bifurcating b-FAD and the distant d-FAD, the site of the weak electron donor (*Demmer et al., 2018*; *Demmer et al., 2017*) thereby creating the B (bifurcation-connected) and D (dehydrogenation-connected) states. The rotating EtfAB shuttle domain joined with the EtfAB base by two mobile linkers is held in a defined trajectory under participation of two segments termed EtfB protrusion and EtfA hairpin (*Demmer et al., 2017*). Some EtfA family members contain an N-terminal Fd domain carrying one or two [4Fe-4S] clusters (*Demmer et al., 2018*) to mediate electron transfer (ET) between the external Fd and the bifurcating FAD. Deduced from the sequence and the signature motif (C(x)$_{17}$CxxCxxC) Ldh-EtfAB possesses a Fd domain with one [4Fe-4S] cluster (*Weghoff et al., 2015*).

In the presented work, a comprehensive analysis of the confurcating Ldh-EtfAB reaction is performed, based on the electron cryo-microscopy (cryo-EM) structure complemented by kinetic analysis of various enzyme variants.

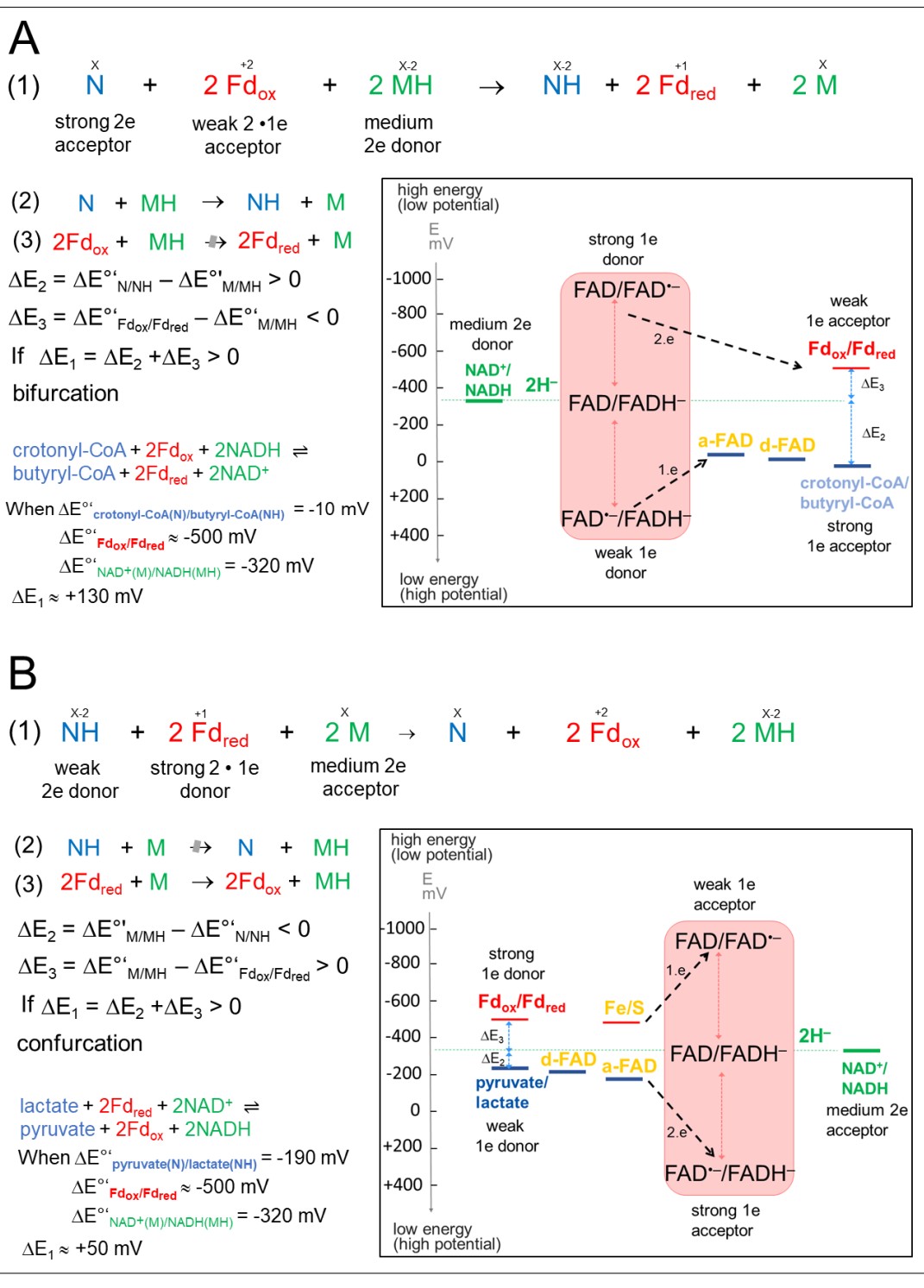

**Figure 1.** Thermodynamic basis. (**A**) The FBEB reaction (1). It involves two reduction reactions (2+3) using the same electron donor (MH) of medium redox potential and two electron acceptors (N and $Fd_{ox}$) of high and low redox potential. Fd can be only replaced by flavodoxin (*Chowdhury et al., 2016*). Biochemical reactions are normally characterized by a pair-wise exchange of electrons between substrates that are protonated in their reduced state. FBEB occur when the positive difference redox potential $\Delta E_2$ between the strong electron acceptor (N) and the medium electron donor (MH) has a higher absolute value than the negative $\Delta E_3$ between $Fd_{ox}$ and MH. In other words, $\Delta E_1$ has to be positive resulting in a negative Gibbs free energy ($\Delta G = -nF\Delta E$, n: moles of electrons exchanged, F: Faraday constant). As example, the electrochemical treatment and the thermodynamic scheme was provided for the reaction of the bifurcating Bcd-EtfAB complex, in which the b-FAD reduced by

*Figure 1 continued on next page*

*Figure 1 continued*

NADH endergonically donates one electron *via* a-FAD to d-FAD and then exergonically one electron to Fd$_{ox}$. (**B**) The reverse FBEC reaction (1) calalyzing two oxidation reaction (2+3) with the same electron acceptor (M). Here, positive $\Delta E_3$ has a higher absolute value than the negative $\Delta E_2$. In the Ldh-EtfAB reaction low-potential Fd$_{red}$ donates an electron to b-FAD *via* one [4Fe-4S] cluster in an endergonic reaction that is driven by the exergonic ET from d-FAD, *via* a-FAD to b-FAD•$^-$. The generated b-FADH$^-$ transfers a hydride to NAD$^+$. For the FBEB/FBEC process the first uphill ET step to b-FAD is reversed except when instanteneously pulled out by the second downhill ET step termed as escapement-type mechanism (*Baymann et al., 2018*). This tight coupling implicates catalytic inactivity in the absence of one of the three substrates and prevention of short circuit reactions e.g. from b-FAD•$^-$ to a-FAD•$^-$ and undesirable side reactions. of the highly reactive b-FAD•$^-$.

The online version of this article includes the following figure supplement(s) for figure 1:

**Figure supplement 1.** The four structurally charcterized EtfB family members; functional units of Bcd-EtfAB (**A**), CarCDE (**B**), FixCX-EtfAB (**C**) and for comparison Ldh-EtfAB (**D**).

## Results and discussion

### Heterologous production, purification and initial characterization of the Ldh-EtfAB complex from *A. woodii*

For structural and functional analysis, the encoding strep-tagged *lctBCD* genes from *A. woodii* were cloned into the expression vector *pET21a* and expressed in the *Escherichia coli* strain BL21(DE3)*ΔiscR* (*Supplementary file 1*). The Ldh-EtfAB complex was purified under anoxic conditions by strep-tactin affinity and Superdex 200 size exclusion chromatography. A denaturing gel showed three

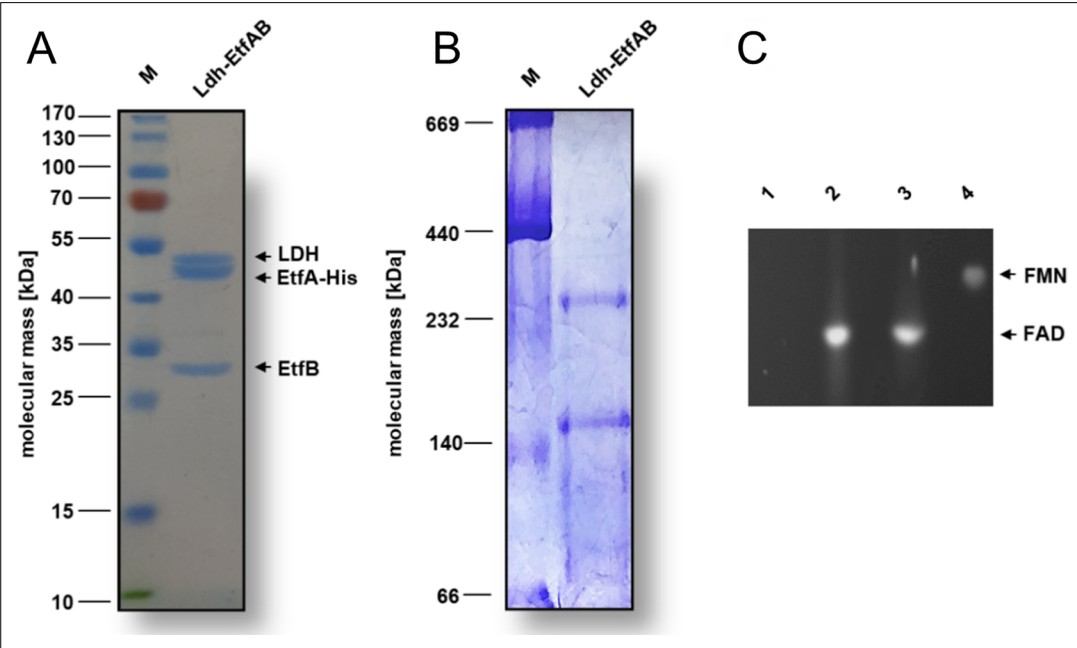

**Figure 2.** Biochemical analysis of the purified Ldh-EtfAB complex. The Ldh-EtfAB complex was anaerobically purified by strep-tactin affinity and Superdex 200 size exclusion chromatography. (**A**) 12% SDS-PAGE and (**B**) native PAGE using 5 µg or 20 µg of the purified protein for separation, respectively. Coomassie Brilliant Blue G250 was applied for protein staining. (**C**) Flavin determination. Flavins of ~1 nmol Ldh-EtfAB were separated on a TLC plate using 60% [v/v] n-butanol, 15% [v/v] glacial acetic acid and 25% [v/v] H$_2$O as the mobile phase. 1 nmol of FAD and FMN were used as standards. For this experiment Ldh-EtfAB was purified with buffers additionally mixed with 5 µM FAD and FMN. Flavins were visualized under UV light. lane 1, buffer; lane 2, Ldh-EtfAB; lane 3, FAD; lane 4, FMN; M, Prestained PageRuler.

The online version of this article includes the following source data and figure supplement(s) for figure 2:

**Source data 1.** SDS-PAGE, native PAGE gels and TLC plate for *Figure 2*.

**Figure supplement 1.** Disassembly of the Ldh-EtfAB complex.

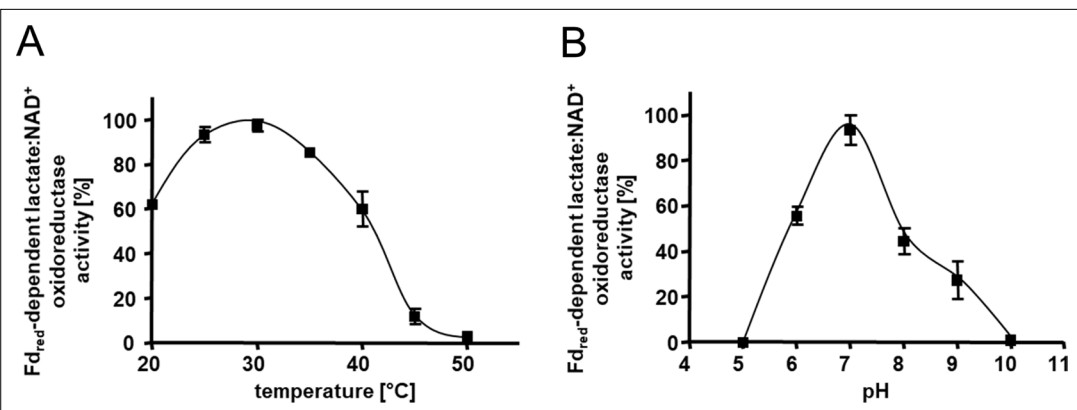

**Figure 3.** Kinetic characterization of the Ldh-EtfAB complex. (**A**) The temperature- and (**B**) the pH- dependency of the FBEC activity. The assay contained 30 μM $Fd_{ox}$, 17 μg carbon monoxide dehydrogenase (CODH); isolated from *A. woodii* (**Hess et al., 2013**), 4 mM $NAD^+$ and 5 μg LDH-EtfAB in a 100% CO gas atmosphere. After 5 min incubation at the appropriate temperature, the reaction was started by adding 50 mM D/L-lactate. The buffer used for measuring the temperature optima was 50 mM Bis-Tris, 50 mM $CaCl_2$, pH 7.0. The buffer used for pH optima determination contained 50 mM MES, 50 mM CHES, 50 mM CAPS, 50 mM Bis-Tris, 50 mM Tris, and 50 mM $CaCl_2$. The reduction of $NAD^+$ was measured by absorption spectroscopy at 340 nm. The data represents the mean and standard deviation of two independent experiments (n=2), each performed in triplicate.

distinct proteins with apparent molecular masses of 51 kDa (LDH), 46 kDa (EtfA), and 29 kDa (EtfB) (*Figure 2A*) that matches to the predicted gene masses of 51.2 kDa, 46.2 kDa, and 29.1 kDa. Native gel electrophoresis and analytical gel filtration data revealed a rather fragile multimeric complex with maximal molecular masses of ≈150 kDa or ≈265 kDa (*Figure 2B*) and ≈250 kDa, respectively. Gel filtration profiles clearly indicate the partial dissociation of the Ldh-EtfAB complex into Ldh and EtfAB (*Figure 2—figure supplement 1*).

The heterologously produced Ldh-EtfAB complex shows similar biochemical properties as the protein isolated directly from *A. woodii* (*Weghoff et al., 2015*). It catalyzes the confurcating $Fd_{red}$-dependent lactate:$NAD^+$ oxidoreductase reaction with a rate of 12.3±1.8 U/mg and the bifurcating pyruvate-dependent NADH:$Fd_{ox}$ oxidoreductase reaction with a rate of 1.2±0.2 U/mg. FBEC activity was optimal between 25°C and 30°C (*Figure 3A*). It decreased below 25 or above 30°C by 40 or 20% and at 45°C to minor values. The Ldh-EtfAB complex had the highest FBEC activity at pH 7 (*Figure 3B*). A decrease of the FBEC activity to 50% could be observed at pH 6 and 8 and inactivity at pH 5 and 10. Similar FBEC activities could be measured, when separately produced Ldh (data not shown) and EtfAB (separated in the gel filtration column) were added together, which demonstrate the stability of the individual components and their capability to assemble.

## Global structural features of the Ldh-EtfAB complex

The cryo-EM structure of the fragile Ldh-EtfAB complex from *A. woodii* was determined from a sample cross-linked with bis(sulfosuccinimidyl)suberate (BS³), because multiple attempts without cross-linking failed. From 9,788 micrographs collected and 674,283 particles extracted a twofold-averaged density map at a mean resolution of 2.43 Å was calculated (*Figure 4A*, *Figure 4—figure supplement 1*, *Table 1*) using Relion (*Scheres, 2012*). The local resolution ranges from 2.3 Å for the Ldh core to 3.1 Å for a peripheral EtfA region (*Figure 4B*). The EM density was only disordered for the N-terminal Fd domain (2-65) of EtfA, for which model building was impossible. The polypeptide chain of all other structural parts could be largely traced with ARP/WARP (*Langer et al., 2008*). Missing residues were afterwards manually incorporated with COOT (*Emsley and Cowtan, 2004*). In particular, the cryo-EM density at the flavin binding sites is highly reliable (*Figure 4—figure supplement 2*).

The Ldh-EtfAB complex was found as a heterohexamer composed of a Ldh dimer forming the core and two EtfAB modules peripherally associated. Exclusively, the Ldh subunits provide the interface between the two Ldh-EtfAB protomers (*Figure 4C*). The active site regions of the two Ldh-EtfAB protomers are ca. 30 Å apart from each other and considered to operate independently. The obtained molecular mass of ca. 255 kDa complex agrees with the value derived from gel filtration and

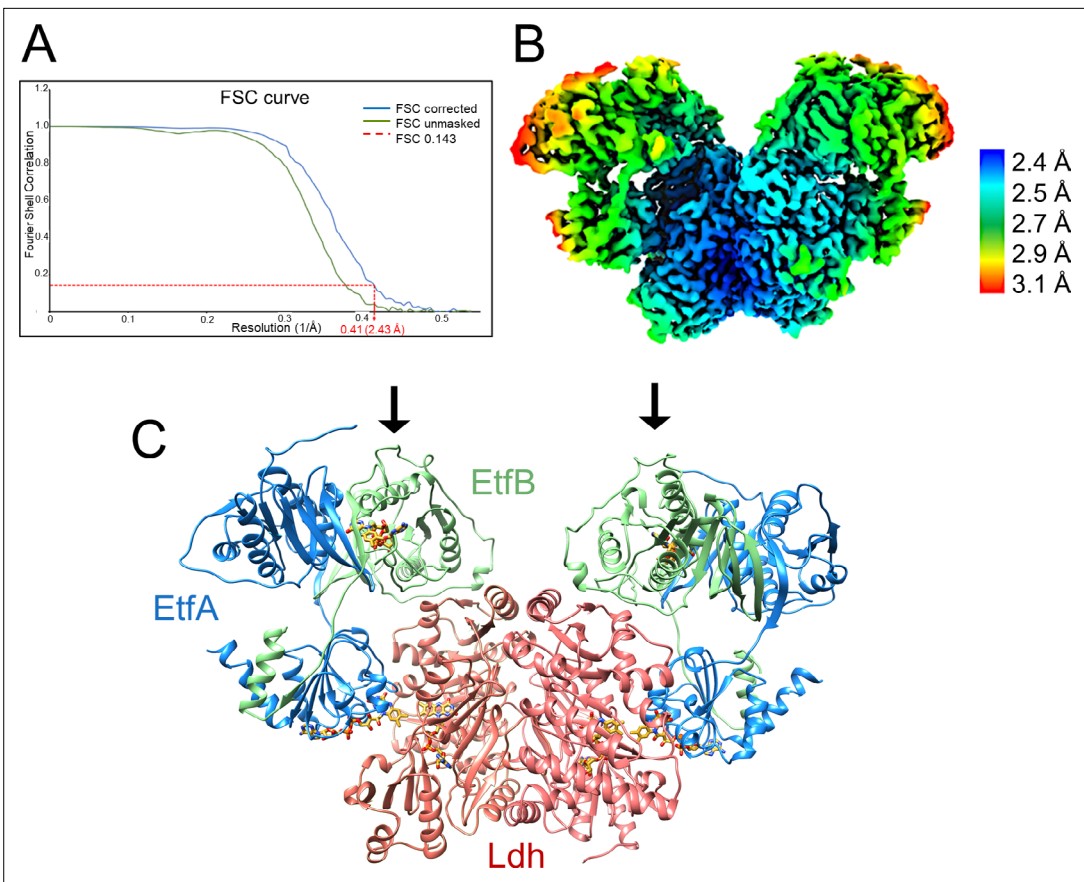

**Figure 4.** Structure of (Ldh-EtfAB)$_2$ complex in the D-state. (**A**) Fourier Shell Correlation curve of unmasked and corrected 3D maps at 0.143 'gold standard' (**B**) Cryo-EM map at 2.43 Å resolution. The density is colored according to the local resolutions. (**C**) Ribbon representation. The Ldh dimer is drawn in red, EtfA in blue and EtfB in green. The 255 kDa heterohexameric complex has an overall size of 150 Å x 100 Å x 60 Å. The docking sites of the two Fd-like domains are marked with black arrows.

The online version of this article includes the following figure supplement(s) for figure 4:

**Figure supplement 1.** Electron microscopy (EM) data.

**Figure supplement 2.** Cryo-EM density at 2.43 Å resolution in the surrounding of a-FAD (**A**), b-FAD (**B**) and d-FAD (**C**).

native PAGE analysis. In accordance with the X-ray structures of the Bcd-EtfAB/CarCDE/Fix-EtfABCX complexes (*Demmer et al., 2018*; *Demmer et al., 2017*; *Feng et al., 2021*) the cryo-EM (Ldh-EtfAB)$_2$ structure shows both Ldh-EtfAB protomers in the resting D (dehydrogenation-connected) state (*Figure 5A*), in which electrons are exchangeable between a-FAD and d-FAD. During the reaction cycle ET has also to be adjusted between a-FAD and b-FAD implicating a rotation of the EtfAB shuttle domain toward the B (bifurcation-connected) state. Analysis of the EM micrographs with the aim to find a small population in the B-state failed. So far, bifurcating oxidoreductase-EtfAB complexes are only experimentally trapped in the D-state therefore considered as stable resting conformation. In contrast, non-bifurcating oxidoreductase-EtfAB complexes form a transient D-like state, which is beneficial as the EtfAB shuttle domain assembles with different dehydrogenase partners (*Leys et al., 2003*). In bifurcating oxidoreductase-EtfAB complexes the B-state appears to be short-living and experimentally difficult to trap. However, a B-state of the Ldh-EtfAB complex was, unexpectedly, found (*Figure 5B*, *Figure 5—figure supplement 2*) by applying the EtfA sequence to the recently published Alphafold2 (*Jumper et al., 2021*) and aligning the the N-terminal domain of EtfA (domain 1) with the cryo-EM Ldh-EtfAB structure (*Figure 5B*). The EtfAB shuttle domain swings ca. 75° from the D-state into a B-state position (*Figure 5—video 1*) that deviates from the previously proposed one (*Demmer et al., 2017*).

**Table 1.** Cryo-EM statistics.

**Experimental data**

| | |
|---|---|
| Protein | Ldh-EtfAB complex |
| State | Resting D-state |

Data collection and Processing

| | |
|---|---|
| Microscope | Thermo Scientific Titan Krios G3i |
| Voltage (kV) | 300 |
| Camera | Gatan K3 summit |
| Exposure time (s) | 6.52 |
| Total dose (e$^-$/Å$^2$) | 106.17 |
| Dose per frame (e$^-$/Å$^2$) | 1.01 |
| Defocus range (μm) | 1.2–2.1 |
| Pizel size (Å) (calibrated) | 0.837 |
| Magnification (nominal) | 105,000 x |
| Symmetry imposed | C2 |
| No. of micrographs | 9,788 |
| Initial particle number | 5,517,853 |
| Final particle number | 674,283 |
| Map resolution (Å) | 2.43 |
| FSC threshold | 0.143 |

Refinement

| | |
|---|---|
| Map-sharpening B factor (Å$^2$) | –67 |
| Model composition | |
| Chains | 6 |
| Protein (Residues) | 2,140 |
| Ligands | FAD: 6; FE: 2 |
| RMSD Bond Length (Å) | 0.005 |
| RMSD Bond Angles (°) | 1.17 |
| MolProbity score | 2.77 |
| Clash score | 4.39 |
| Ramachandran plot (%) | |
| Favored | 97.0 |
| Allowed | 2.9 |
| Outliers | 0.1 |
| Rotamer outliers (%) | 1.9 |
| ADP (B-factor) (min/max/mean) | |
| Protein | 55.1/151.3/90.6 |
| Ligand | 60.6/93.4/79.2 |

The oscillation of the EtfAB shuttle domain between the D- and B-states is realized by a fixed interface between the rigid Ldh core and EtfAB base platforms and a variable interface either between the EtfAB shuttle domain and Ldh or between the EtfAB shuttle domain and the EtfAB base (*Figure 5*). The fixed interface is constituted by the small segment 185–193 of EtfB composed of an elongated loop and a short helix which is well conserved in bifurcating and non-bifurcating EtfBs and referred to as recognition loop (*Figure 5—figure supplement 1A*; *Toogood et al., 2007*). The variable interface in the D-state is centered around a-FAD and d-FAD. In the Ldh-EtfAB complex the nonpolar xylene moiety of the isoalloxazine rings point towards each other; the edge-to-edge distance is 3.7 Å (*Figure 5—figure supplement 1B*). For adjusting a productive interflavin distance the contact region of the D-state has to be individually adapted for each dehydrogenase /oxidoreductase-EtfAB complex. A special situation is reported for the FixCX-EtfAB complex, as ET between the EtfAB shuttle domain and FixC is mediated by the additional Fd subunit FixX (*Feng et al., 2021*). In the variable interface of the B-state a-FAD becomes attached to the elongated loop linking β-strands 201:206 and 213:217 of EtfA and the EtfB protrusion (*Figure 5—figure supplement 1C*). The xylene rings of b-FAD and a-FAD point to each other and the shortest distance between their methyl groups is 11.8 Å.

## The Ldh subunit and its active site

Ldh is built up of two domains, an N-terminal FAD domain (1-218) subdivided into two α + β subdomains, a cap domain (219-417) characterized by an antiparallel β-sheet element and an extended C-terminal arm (418-467) attached to the FAD domain (*Figure 6A*). This fold classifies Ldh as a member of a flavoenzyme family with p-cresol methyl hydroxylase (CMH) (*Cunane et al., 2000*), vanillyl-alcohol oxidase (*Mattevi et al., 1997*), MurB (*Benson et al., 1997*) and membrane-associated Ldh (*Dym et al., 2000*) as prototypes (*Fraaije et al., 1998*); their corresponding rms deviations from Ldh of *A. woodii* are 2.6 Å (1DIQ, 407 of 467 residues used), 2.6 Å (1AHU, 399 of 467), 3.5 Å (2MBR, 203 of 467) and 2.4 Å (1FOX, 371 of 467). Except for the membrane-associated Ldh CMH-type flavoenzymes are homodimers. D-FAD is embedded between the two α + β subdomains except for the isoalloxazine ring that protrudes beyond

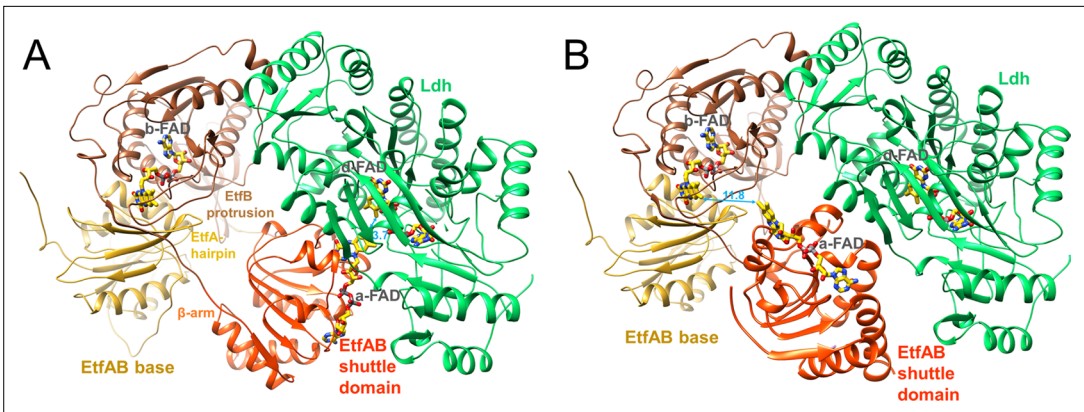

**Figure 5.** Structural states of the (Ldh-EtfAB)₂ complex. (**A**) The experimentally characterized D-state. The EtfAB shuttle domain (red) is spatially separated from the rigid EtfAB base (EtfA domain I in gold and EtfB domain III in brown) and attached to Ldh (green). a-FAD and d-FAD are 3.7 Å, a-FAD and b-FAD ca. 37 Å apart from each other. (**B**) The B-state. By combining the experimental EM data and Alphafold2 calculations a plausible B-state could be established for the first time. Starting from the D-state the EtfAB shuttle domain swings 75° to adjust an ET distance between a-FAD and b-FAD. The shortest distance between the isoalloxazine rings is 11.8 Å. Upon rotation the contact to Ldh is largely lost while that to the EtfAB base is formed.

The online version of this article includes the following video and figure supplement(s) for figure 5:

**Figure supplement 1.** Interfaces between Ldh, the EtfAB base and the EtfAB shuttle domain.

**Figure supplement 2.** Quality of the AlphaFold2 EtfA model.

**Figure 5—video 1.** Transition between the D- and B-states.

https://elifesciences.org/articles/77095/figures#fig5video1

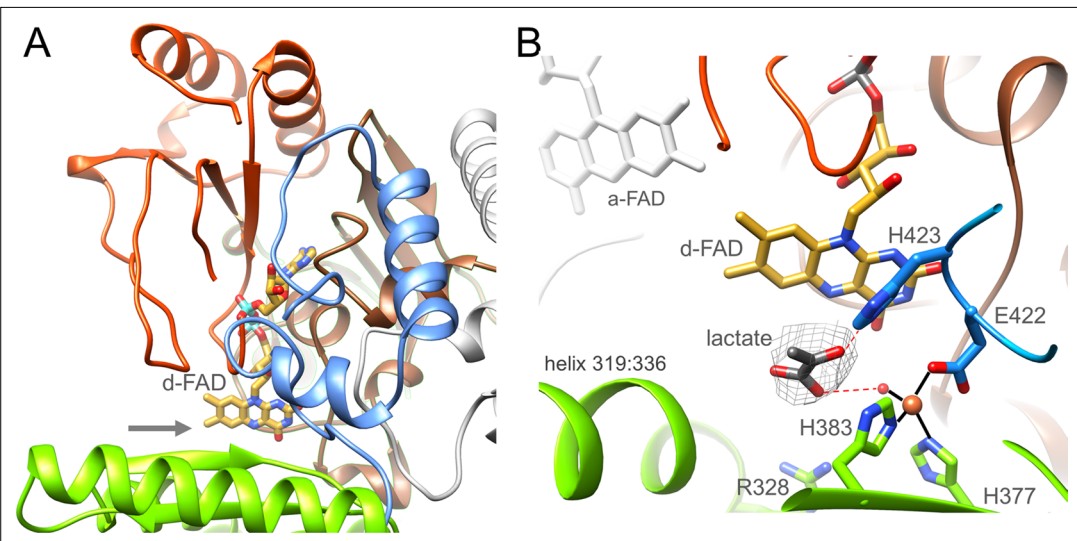

**Figure 6.** Ldh. (**A**) The overall structure. The Ldh subunit belongs to the CMH flavoenzyme family composed of two α + β subdomains (orange-red and brown), a cap domain (green) and a C-terminal extension (blue). d-FAD (as stick with carbon in gold) is positioned between them. The isoalloxazine ring is fixed by several van der Waals contacts and two hydrogen-bonds between FAD N3C2O and Gly153CONH and between FAD O4 and Gly138NH of Ldh. The entrance of lactate is marked by a grey arrow. (**B**) The active site. Lactate was tentatively modeled into a density (gray) in front of the N5 of d-FAD involving a metal site for its binding. Its hydroxy group appears to be activated by His423. We tentatively assigned the metal as $Fe^{2+}$ (orange) due to its addition during purification.

The online version of this article includes the following source data for figure 6:

**Source data 1.** Sequence alignment of Ldh.

**Table 2.** Specific activities of recombinant Ldh-EtfAB and site-specifically changed variants with artificial electron donors/acceptors.

| | WT | ΔFe/S | ΔFe/S-arm | D189A | R205A | Δb-FAD | ΔNAD⁺ | ΔSPT |
|---|---|---|---|---|---|---|---|---|
| Fd$_{red}$:lactate: NAD⁺ (340 nm) | 12.3±2.3 | 0.1±0.05 | 0.1±0.06 | 7,7±2.6 | 0.01±0.01 | 0.9±0.3 | 0.2±0.1 | 1.1±0.3 |
| NADH:pyruvate:Fd$_{ox}$ (340 nm) | 1.2±0.2 | 0.5±0.03 | 0.1±0.1 | 0.9±0.1 | 0.02±0.02 | 0.06±0.02 | 0.01±0.01 | 0.01±0.05 |
| lactate:K₃[Fe(CN)₆] (420 nm) | 19.0±5.2 | 13.9±5.4 | 14.6±4.8 | 11.1±6.7 | 15.8±4.3 | 8.3±9.2 | 8.5±7.3 | 8.7±6.5 |
| NADH:DCPIP (600 nm) | 8.8±2.5 | 7.3±0.2 | 9.7±1.5 | 5.6±2.5 | 6.3±2.3 | 0.4±0.2 | 0.2±0.1 | 9.2±0.4 |

NAD⁺, Fd, ferricyanide or DCPIP reduction was measured at 340, 430, 450 or 600 nm, respectively. Values are given as specific activities in U/mg protein.

the FAD domain towards the cap domain. Therefore, the isoalloxazine is well accessible from bulk solvent (*Figure 6A*).

In front of its *si*-side the substrate binding site can be easily reached by lactate/pyruvate *via* a wide gate (*Figure 6A*) framed by Gly138, Met155, Gly329, and Leu332. The substrate binding sites is essentially formed by Leu80, Arg328, Leu332, Glu344, Asp346, His377, His384, Tyr386, Glu422, and His423 directly or indirectly involved in lactate binding/oxidation (*Figure 6B*) and well conserved in the membrane-associated Ldh (*Figure 6—source data 1*; *Dym et al., 2000*). Notably, we detected a putative metal binding site in the substrate binding cavity ligated by His377, His384, Glu422 and a water molecule. An unexplained density in front of d-FAD corresponds to the profile of pyruvate or lactate (*Figure 6B*). Lactate was modeled in a manner that the hydride transferring carbon is 3.6 Å apart from N5 and the hydroxy group is hydrogen-bonded with the invariant His423 (*Figure 6— source data 1*) perhaps acting as a catalytic base for abstracting the hydroxy proton during hydride transfer from C2 to N5 of FAD. The carboxyl group of the lactate would be fixed by interactions with the metal-ligating water and Arg328.

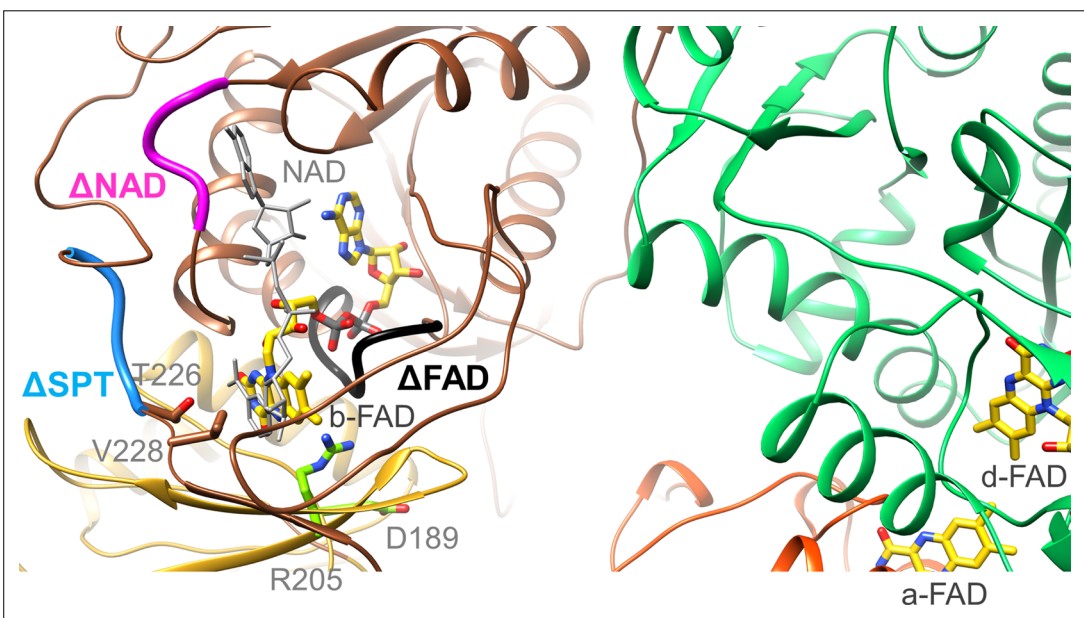

**Figure 7.** Location of the EtfA and EtfB mutations. The highly conserved segments [122]DGDTAQVGP[130], [87]RXFXG[91] and [223]SPT[225] of the Δb-FAD, ΔNAD⁺ and ΔSPT variants are highlighted in blacks, magenta and blue. The positions of R205 and D189 are shown as ball-and-sticks. NAD⁺ was modeled as gray stick.

The online version of this article includes the following source data and figure supplement(s) for figure 7:

**Figure supplement 1.** SDS-PAGE of the site-specifically changed Ldh-EtfAB variants. The Ldh-EtfAB variants were anaerobically purified by strep-tactin affinity and Superdex 200 size exclusion chromatography.

**Figure supplement 1—source data 1.** SDS-PAGE gels for *Figure 7—figure supplement 1*.

## The EtfAB module carrying b-FAD

The bifurcating EtfAB module of the (Ldh-EtfAB)$_2$ complex is structurally related to other family members (*Figure 1—figure supplement 1*) reflected in rms deviations of 1.4 Å (404 of 443 residues), 1.4 Å (416 of 443) and 2.4 Å (368 of 443) between its EtfAB base (domain I, 65–269 and domain III, 2–236) and those of the Bcd-EtfAB (5OL2, *Demmer et al., 2017*), CarCDE (6FAH, *Demmer et al., 2018*), and Fix-EtfABX (7KOE, *Feng et al., 2021*). The corresponding values for the EtfAB shuttle domain (EtfA 270–398, EtfB: 237–265) are 1.4 Å (155 of 158), 1.8 Å (153 of 158), and 2.1 Å (112 of 158), respectively. b-FAD is considered as the heart of FBEB/FBEC and exchanges electrons with the three spatially separated redox carriers a-FAD, the [4Fe-4S] cluster and NAD$^+$ (*Chowdhury et al., 2014*). For evaluating its binding and catalytic function, crucial residues were substituted by the single-nucleotide exchange method *via* corresponding primers previously described (*Supplementary file 1*; *Demmer et al., 2018*). For all enzyme variants presented (*Table 2*) the yield and subunit composition correspond to that of the wild-type enzyme (*Figure 7—figure supplement 1*). Bifurcating oxidoreductase-EtfAB complexes including Ldh-EtfAB contain the conserved $^{122}$DGDTAQVGP$^{130}$ stretch as recognition motif for b-FAD binding (*Figure 7*), which is absent in non-bifurcating Etfs (*Buckel and Thauer, 2018b*; *Chowdhury et al., 2014*; *Garcia Costas et al., 2017*). The [Δb-FAD] variant consisting of the D122A, D124A, T125G, Q127G, V128A and P130A substitutions shows an amount of 1.7±0.4 mol FAD per mol Ldh-EtfAB compared with 2.9±0.3 mol FAD per mol wildtype Ldh-EtfAB. The FBEC and FBEB monitoring Fd$_{red}$-dependent lactate:NAD$^+$ and pyruvate-dependent NADH:Fd$_{ox}$ oxidoreductase activities were reduced to 7% and not anymore measurable, respectively. Likewise, the NADH:DCPIP oxidoreductase activity, measuring the hydride transfer from NADH to b-FAD, was abolished. Both findings are referable to the small amount of b-FAD bound. On the other hand, lactate to pyruvate oxidation with ferricyanide as electron acceptor (located at the remote d-FAD, *Figure 7*) is only reduced by a factor of 2 in the [Δb-FAD] variant (*Table 2*) indicating two independently acting active sites.

Moreover, we exchanged the conserved residue R205 of EtfA (*Figure 7*) that forms a hydrogen-bond to N5 of b-FAD. As found for the equivalent CarCDE variant (*Demmer et al., 2018*) the R205A variant of Ldh-EtfAB completely losts the capability of FBEC/FBEB (*Table 2*). Although all analyzed FBEB enzymes share a positively charged arginine/lysine residue (*Chowdhury et al., 2014*; *Demmer et al., 2015*; *Lubner et al., 2017*; *Wagner et al., 2017*; *Watanabe et al., 2021*) in contact with N5 of b-FAD, its specific function beyond flavin binding is still obscure (*Kayastha et al., 2021*) albeit an effect for stabilizing FAD•$^-$ by the positive charge was reported recently (*Mohamed-Raseek and Miller, 2022*). The maintained NADH:DCPIP oxidoreductase activity excludes larger rearrangements of the binding site of b-FAD and thus supports the crucial importance of Arg205 for its unusual redox behavior. D189 of EtfA is 11 Å apart from b-FAD (*Figure 7*) and in the B-state 4.5 and 10 Å apart from the previously proposed and AlphaFold2-calculated a-FAD position, respectively (*Demmer et al., 2017*; *Jumper et al., 2021*). The moderate 35% and 15% decrease of the FBEC and FCEC activities measured for the D189A variant rather argues for a FAD binding site farther from D189 and thus for the AlphaFold2-derived B-state. As a control R205A and D189A variants were still capable to reduce DCPIP with NADH or ferricyanide with lactate (*Table 2*).

To substantiate the assumed binding site for NAD$^+$/NADH (*Chowdhury et al., 2014*), we prepared the [ΔNAD$^+$] variant by substituting R87, F89 and G91 to alanine (*Figure 7*). NADH oxidation in the FBEC activity assay was reduced to 8% of the wildtype and the NAD$^+$ reduction in the FBEB activity assay was no longer measurable. In addition, DCPIP could not be reduced by NADH anymore but ferricyanide with lactate with a twofold lower activity (8.5±7.3 U/mg; *Table 2*). Altogether, the vital role of b-FAD as bifurcating flavin and of NADH as its hydride donor/acceptor was demonstrated for the EtfB family independent from structural and electrochemical interpretations (*Chowdhury et al., 2014*; *Demmer et al., 2018*; *Demmer et al., 2017*; *Feng et al., 2021*; *Sucharitakul et al., 2020*).

Finally, we exchanged the highly conserved residues S223, P224, and T225 localized at the anchor of the EtfB arm in the vicinity of the NAD$^+$ binding site (*Figure 7*). This [ΔSPT] variant showed 90% lower Fd$_{red}$-dependent lactate:NAD$^+$ and almost completely lost pyruvate-dependent NADH:Fd$_{ox}$ oxidoreductase activities (*Table 2*). The DCPIP: NADH and lactate:K$_3$Fe(CN)$_6$ oxidoreductase activities were 50% lower than those of the wild-type enzyme indicating that these mutations primarily do not affect the individual redox reactions but their coupling. In the context of previous data (*Demmer et al., 2018*; *Demmer et al., 2017*; *Schut et al., 2019*), the anchor at the EtfB arm influences the transmission of NAD$^+$ binding towards the EtfB arm and thus the oscillation of the EtfAB shuttle

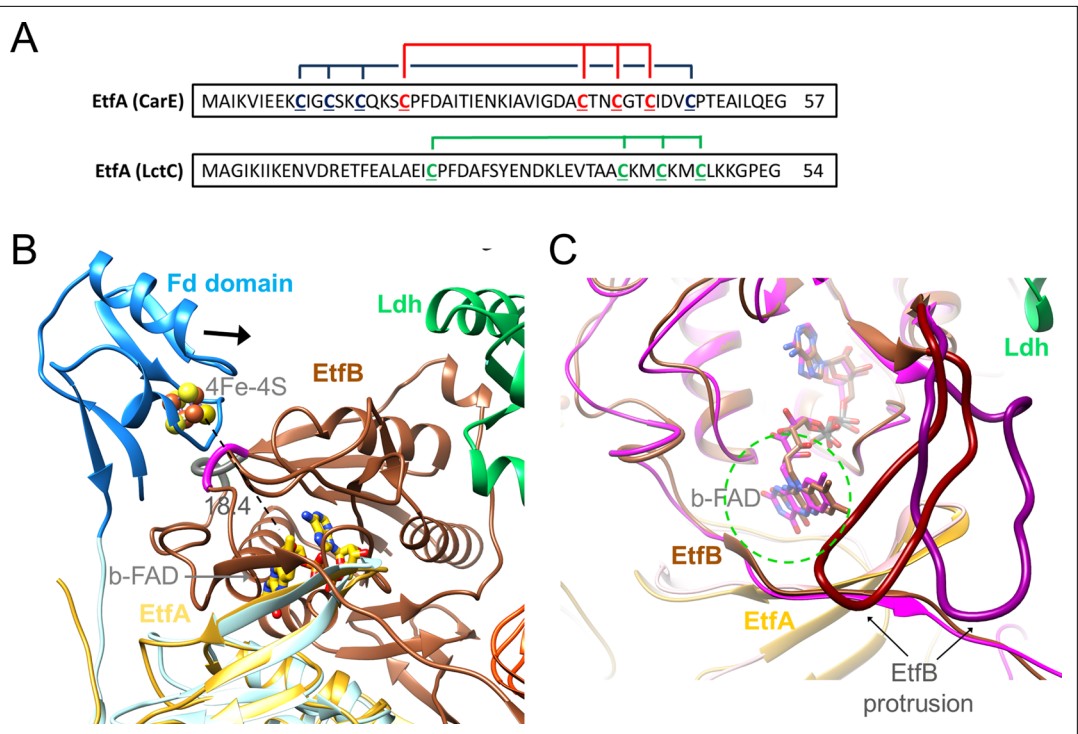

**Figure 8.** Ferredoxin (Fd) domain. (**A**) The sequence of the N-terminal Fd domain of the CarCDE and the Ldh-EtfAB complexes. EtfA of the Ldh-EtfAB complex contains one [4Fe-4S] cluster ligated with four cysteines marked in green. In comparison, the equivalent CarE of the CarCDE complex possesses two [4Fe-4S] clusters marked in blue and red. The latter is shared between Ldh-EtfAB and CarCDE. (**B**) The cryo-EM structure of the Ldh-EtfAB complex (EtfA, lightorange; EtfB, brown; Ldh, green) superimposed with EtfA (lightblue) including the Fd domain (dodger blue) calculated by AlphaFold2. The Fd domain interacts with EtfB *via* the segment (gray) linking the anchor of the EtfB arm (225-232) and the terminal strand 203:207 of the central β-sheet and *via* the loop (magenta) following strand 81:85. The [4Fe-4S] cluster (in ball-and-stick) of the Fd domain was modeled based on the properly positioned four cysteine sulfurs. For productive ET the Fd domain moves toward EtfB (black arrow). (**C**) Accessibility of b-FAD. b-FAD is more shielded in Ldh-EtfAB (colors as in A) than in the superimposed CarCDE (hotpink) due to the reoriented EtfB protrusion such that an arriving external Fd might not find a docking site sufficiently close to b-FAD. The approximate diameter of Fd is drawn as dashed green circle.

domain (*Figure 5*). This finding contributes to the understanding of the orchestration of the complex FBEC/FBEB events upon substrate binding.

## The Fd domain

EtfA of the Ldh-EtfAB complex contains a Fd domain with one [4Fe-4S] cluster (*Weghoff et al., 2015*) in contrast to the structurally characterized EtfA subunits of Bcd-EtfAB and CarCDE endowed with no N-terminal Fd or an N-terminal Fd domain with two [4Fe-4S] clusters, respectively (*Figure 8A*; *Chowdhury et al., 2014*; *Demmer et al., 2018*). In the cryo-EM structure the density of the Fd domain in both Ldh-EtfAB protomers is highly disordered such that chain tracing and also the detection of the [4Fe-4S] cluster is not feasible. Its location could, however, be identified close to the position of the EtfA model determined by AlphaFold2 (*Figure 8* and *Figure 5—figure supplement 2*; *Jumper et al., 2021*). Accordingly, the preserved [4Fe-4S] cluster in the Ldh-EtfAB complex is the one closer to b-FAD when compared with superimposed CarCDE (*Demmer et al., 2018*). Its distance of 18 Å to b-FAD is still a bit too long for effective ET. However, the highly mobile Fd domain is shifted towards EtfB reaching a suitable ET distance between the [4Fe-4S] cluster and b-FAD when using the EtfA and EtfB sequences stringed together for AlphaFold2 calculations (*Figure 8B*).

For functional analysis, the entire Fd domain (2–65) was truncated or cysteines 41, 44, and 47 coordinating the [4Fe-4S] clusters were exchanged to alanine. In these [ΔFe/S-domain] and [ΔFe/S] variants, the irons are completely lost (0.3±0.2 and 0.5±0.3 mol Fe/mol protein); the stability and

subunit composition were, however, not affected (*Figure 7—figure supplement 1*) and the overall yield increases up to fourfold compared to the complete Ldh-EtfAB complex. Kinetic analysis revealed that the FBEC/FBEB activities were abolished in both variants (*Table 2*), whereas the functionality of the isolated active sites is maintained. The NADH:DCPIP oxidoreductase activity of the [ΔFe/S] or [ΔFe/S-arm] variants was 7.3±0.2 and 9.7±1.5 U/mg, respectively, and the lactate:K$_3$Fe(CN)$_6$ oxidoreductase activity 13.9±5.4 and 14.6±4.8 U/mg, respectively (*Table 2*). In contrast, the CarCDE complex preserves the FBEB capability after cutting off its Fd domain. We, therefore, conclude that external Fd can be placed next to b-FAD in truncated CarCDE but not in the [ΔFe/S] or [ΔFe/S-arm] variants of Ldh-EtfAB. Superimposed Ldh-EtfAB and CarCDE reveals a displacement of the EtfB protrusion by nearly 15 Å, which might impair Fd binding close to b-FAD in Ldh-EtfAB (*Figure 8C*). In summary, this result shows the vital importance of the [4Fe-4S] cluster in Ldh-EtfAB as specific adaptor for external Fd binding. It is worth mentioning that the non-bifurcating EtfABs from *Clostridium propionicum* (Cprop2325) highly related to their bifurcating partners contain an insertion region that may block Fd

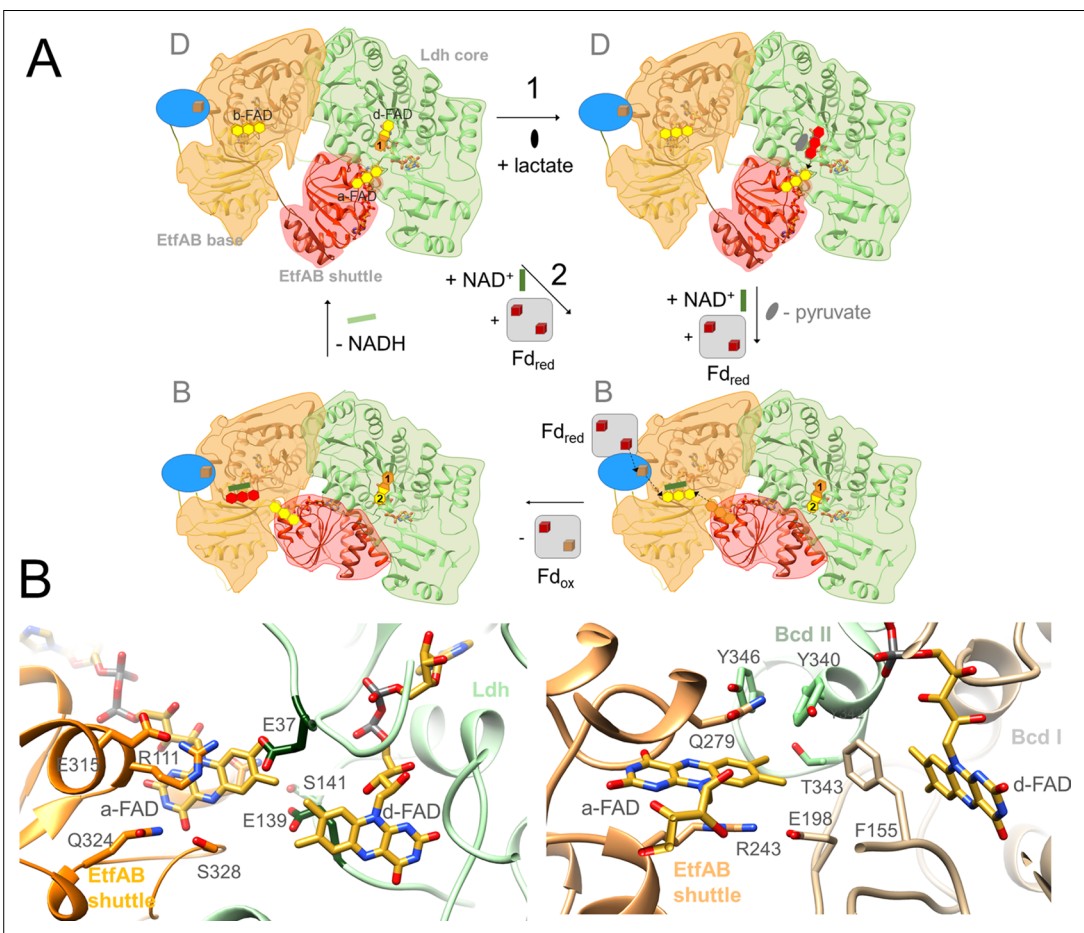

**Figure 9.** The Ldh-EtfAB reaction. (**A**) The mechanism. The reaction cycle was outlined in the direction of the thermodynamically preferred confurcation direction. It starts from the structurally characterized D-state and runs through two rounds (termed 1+2). The three-membered rings of FAD, FAD•⁻ / FADH• and FADH⁻ are drawn in yellow, orange and red. The d-FAD isoalloxazine is shown with orange and yellow moieties representing in the 1. round FADH• (FAD•⁻) and in the 2. round FAD. The Fd domain is drawn as blue ellipse and the external Fd as grey box. Ldh-EtfAB is normally present in the resting, FBEC inactive D-state. A D- to B-state transition upon 75° rotation of the EtfAB shuttle domain and an induced-fit event upon Fd binding momentary build up the acitive site for firing two electrons from two directions to b-FAD (right-bottom panel). (**B**) Microenvironments of a-FAD and d-FAD in the D-state of Ldh-EtfAB (left panel) and Bcd-EtfAB (right panel). In Ldh-EtfAB the two isoalloxazine rings are closer and the protein surrounding is significantly different compared to Bcd-EtfAB/CarCDE. Glu37 and Glu139 (dark green) contacting a-FAD and d-FAD may destabilize an electron-rich reduced state resulting in a lower redox potential (*Figure 1B*).

binding, as already predicted (*Buckel and Thauer, 2018b*), in a manner related to the EtfB protrusion of Ldh-EtfAB.

## Enzymatic mechanism

Kinetic studies on the recombinant Ldh-EtfAB complex indicated a 10 times higher enzymatic activity for the FBEC than for the FBEB process. Therefore, the proposed mechanism is outlined in this direction (*Figure 9A*) starting from the D-state. (1) Lactate binds into a pocket in front of the isoalloxazine of d-FAD and is oxidized to pyruvate *via* a hydride transfer to N5 forming d-FADH⁻ (*Figure 6*). The active site is formed by residues conserved in FBEC and membrane-spanning Ldhs (*Dym et al., 2000*; *Figure 6—source data 1*) suggesting a related enzymatic mechanism. (2) One electron of d-FADH⁻ rapidly flows to a-FAD thereby generating d-FADH• (d-FAD•⁻) and a-FAD•⁻ (a-FADH•) (*Figure 5*). It is worth to mention that a-FAD might be already present as flavosemiquinone in the resting state of the cell and would be reduced in this case to a-FADH⁻. (3) NAD⁺ binds into a pocket at the *si*-side of b-FAD as verified by site-directed mutagenesis experiments (*Table 2*, *Figure 7*) in agreement with previous studies (*Chowdhury et al., 2014*). In the characterized D-state Thr226 and Val228 of EtfB occupy the frontside of b-FAD and have to be pushed aside upon NAD⁺ binding (*Figure 7*). According to reports on Bcd-EtfAB, CarCDE and FixCX-EtfAB complexes (*Chowdhury et al., 2014*; *Demmer et al., 2018*; *Schut et al., 2019*) and supported by mutational analysis (*Table 2* and *Figure 7*) NAD⁺ binding substantially influences the conformation of the EtfB anchor and thus also the adjustment of the B- and D-states. (4) External Fd binds to the mobile Fd domain of EtfA, thereby rigidifies the latter and adjusts a productive ET geometry between their two [4Fe-4S] clusters and b-FAD (*Figure 8B*). A related situation was reported for the bEtfB family member CarCDE (*Demmer et al., 2018*). (5) Perhaps induced by NAD⁺ and/or Fd$_{red}$ binding, the EtfAB shuttle domain rotates into the B-state (*Figure 5—video 1*). (6) Fd$_{red}$ fires one electron *via* the [4Fe-4S] cluster of EtfA to the bifurcating b-FAD (*Figures 8B and 9A*). (7) The produced energy-rich b-FAD•⁻ is instantaneously reduced to b-FADH⁻ by a-FAD•⁻ of the EtfAB shuttle domain arrested in the B-state (*Figure 5B*). According to general electrochemical principles of FBEB both one-electron donation processes to b-FAD accompanied by protonation are strictly coupled (*Figure 1B*; *Baymann et al., 2018*; *Nitschke and Russell, 2012*). (8) The nicotinamide ring of NAD⁺ attached parallel to the isoalloxazine ring becomes reduced by transferring a hydride from N5 of b-FADH⁻ to C4 of NAD⁺. (9) Upon release of NADH and/or Fd$_{ox}$ the EtfAB shuttle domain swings back into the D-state and a-FAD uptakes the remaining electron of d-FADH• (d-FAD•⁻). In analogy to the first round b-FAD is reduced in the B-state by one-electron transfers from Fd$_{red}$ and a-FAD•⁻ rotated before from the Ldh to the EtfAB surface (*Figure 5—figure supplement 1*; *Figure 5—video 1*). Finally, a second NAD⁺ is reduced to NADH (*Figure 9A*).

The Ldh-EtfAB complex catalyzes the first dominantly electron-confurcating reaction structurally studied. As the overall direction of the process is thermodynamically driven, the reaction cycle becomes just reversed (*Figure 1*). All EtfB family members share the substrates NAD⁺ and ferredoxin, solely the high-potential substrate varies and determines the direction of the reaction. As mentioned, the redox potential E°′ of the pyruvate/lactate pair is with –190 mV substantially lower than that of the crotonyl-CoA/butyryl-CoA pair of –10 mV, which turns a bifurcation into a confurcation event (*Figure 1B*). For ensuring a smooth ET the redox potentials of the electron carriers a-FAD and d-FAD between b-FAD and lactate should be also lower in Ldh-EtfAB than in Bcd-EtfAB and thus their microenvironment substantially different (*Figure 9B*). More negatively and less positively charged residues qualitatively stabilize, in general, the more electron-deficient oxidized relative to the more electron-rich reduced state and consequently decrease the redox potential. Structural inspections of Ldh-EtfAB, indeed, reveal two acidic residues, Glu37 and Glu139, in van-der-Waals contact with the isoalloxazines of a-FAD and d-FAD, which are conserved in bifurcating Ldh but not in membrane-spanning Ldh and have no equivalent in Bcd-EtfAB (*Figure 9B*). This finding exemplifies that all redox-active cofactors in FBEB/FBEC enzymes have to be specifically adjusted by the surrounding protein matrix to play their role in a finely tuned energy landscape (*Figure 1B*).

## Materials and methods

### Cloning of lctBCD and generation of site-specifically mutated variants

The genes *lctBCD* (3507 bp) were amplified from chromosomal DNA using the primers lctBCD_pET21a_for and lctBCD_pET21a_rev (*Supplementary file 1*). The expression vector *pET21a* (5406 bp) was amplified using the primers pet21a_for and pet21a_rev (*Supplementary file 1*). Afterwards, *lctBCD* and *pET21a* were fused *via* Gibson Assembly (New England Biolabs) and transformed into *E. coli* HB101 (*Supplementary file 1*). A sequence encoding a Strep-tag was introduced at the 3'-end of the gene *lctC* coding for EtfA by using corresponding primers (*Supplementary file 1*). The resulting plasmid *pET21a_lctBC-StrepD* was the template for site-directed mutagenesis. Nucleic acid changes were introduced using corresponding primers at desired loci in the encoding genes (*Supplementary file 1*). Generated plasmids were checked by sequencing. *pET21a* plasmids were transformed into *E. coli* BL21(DE3) Δ*iscR*.

### Production and purification of the Ldh-EtfAB complex in *E. coli* BL21(DE3)Δ*iscR*

*E. coli* BL21(DE3)Δ*iscR* was grown aerobically in modified LB-medium (1% Trypton, 1% NaCl, 0.5% yeast extract, 100 mM MOPS, pH 7.4) supplemented with 4 mM ammonium-iron(II)citrate and 25 mM glucose at 37°C. At an $OD_{600}$ of 0.5–0.8 the culture was transferred into a sterile, anaerobic Müller-Krempel flask (Glasgerätebau Ochs, Bovenden-Lenglern, Germany) and supplemented with 2 mM cysteine and 20 mM fumarate. After closing the culture with a butyl stopper, the flask was further incubated at 16°C. As soon as the culture was cooled down to 16°C gene expression was induced by addition of IPTG to a final concentration of 1 mM. All steps from this point onward were executed in an anaerobic chamber (Coy Laboratory Products, Grass Lake, USA) with a mixed gas phase of $N_2/H_2$ (95:5 [v/v]). After 16–19 hr cells were anaerobically harvested, resuspended in 200 mL buffer W (50 mM Tris, 150 mM NaCl, 20 mM $MgSO_4$, 20% (v/v) glycerol, 4 µM resazurin, 5 µM FAD, 2 mM DTE, pH 8) and disrupted once in a French press (SLM Aminco, SLM Instruments, USA) after addition of DNase I and 0.5 mM PMSF (setting high, 1000 psi). The lysate was centrifuged (14,000 x g, 20 min, RT) to separate undisrupted cells from the crude extract. Subsequently, the Ldh-EtfAB complex was purified by affinity chromatography on Strep-Tactin high-capacity material (IBA Lifesciences GmbH, Göttingen, Germany) using the elution buffer E (buffer W+5 mM desthiobiotin). Fractions containing Ldh-EtfAB were pooled and concentrated *via* ultrafiltration (Vivaspin 6, 30 kDa Cut-off, Sartorius Stedim Biotech GmbH, Göttingen, Germany) to a volume of 500 µL. The concentrated sample was separated with a flow rate of 0.5 mL/min on a Superdex 200 10/300 GL column (GE Healthcare, Little Chalfont, UK), previously equilibrated with buffer W. For cryo-EM studies buffer S (20 mM HEPES, 150 mM NaCl, pH 7.5) was used for the size exclusion step. Afterwards, the Ldh-EtfAB complex was stabilized with the bis(sulfosuccinimidyl)suberate $(BS^3)$ crosslinker (Thermo Fisher Scientific, Waltham, USA) by incubating the sample ($\approx$ 1 mg) with 1 mM $BS^3$ for 20 min at room temperature. The assay was stopped by addition of 50 mM Tris. To get rid of impurities, the crosslinked-sample was again separated with a flow rate of 0.5 mL/min on a Superdex 200 10/300 GL column (GE Healthcare, Little Chalfont, UK), previously equilibrated with buffer S.

### Assays of lactate dehydrogenase activity

Kinetic measurements were routinely performed in a $N_2$ atmosphere at 30°C in 1.8 mL anaerobic cuvettes (Glasgerätebau Ochs), which were sealed with rubber stoppers. Physiological $Fd_{red}$-dependent lactate:$NAD^+$- and pyruvate-dependent NADH:$Fd_{ox}$ oxidoreductase activities are measured as previously described (*Weghoff et al., 2015*). The same assay conditions were used for recording the lactate:$K_3Fe(CN)_6$- and NADH:DCPIP oxidoreductase activities using 1 mM $K_3Fe(CN)_6$ (ferricyanide) and 500 µM DCPIP (dichlorophenol indophenol), respectively. The $NAD^+$/NADH ($\varepsilon$=6.3 mM$^{-1}$ cm$^{-1}$), Fd ($\varepsilon$=13.1 mM$^{-1}$ cm$^{-1}$) purified from *C. pasteurianum* (*Schönheit et al., 1978*), DCPIP ($\varepsilon$=20,7 mM$^{-1}$ cm$^{-1}$) or $K_3Fe(CN)_6$ ($\varepsilon$=1 mM$^{-1}$ cm$^{-1}$) oxidations/reductions were monitored at 340, 430, 600, or 420 nm by UV/Vis spectroscopy, respectively. One unit (U) equals 2 µmol of electrons transferred per min.

## Analytical methods

Protein concentration was measured according to *Bradford, 1976*, The protein complex was separated by PAGE using the SDS buffer system of *Laemmli, 1970* or *Wittig et al., 2007* and stained with Coomassie brilliant blue G250. The molecular mass of the purified Ldh-EtfAB complex was determined using a calibrated Superdex 200 column, buffer E and defined size standards (ovalbumin: 43 kDa; albumin: 158 kDa; catalase: 232 kDa; ferritin: 440 kDa). The iron content of the purified enzyme was determined by colorimetric methods (*Fish, 1988*) and the flavin content by thin layer chromatography (TLC) as described before (*Bertsch et al., 2013*).

## Single-particle electron cryo-microscopy

For sample vitrification C-Flat R1.2/1.3 copper, 300-mesh grids (Electron Microscopy Sciences) were glow discharged thrice with a PELCO easiGlow device at 15 mA for 45 s. Four µl of protein with a concentration of 1.8 mg/ml were applied to a freshly glow-charged grid, blotted for seconds at 4 °C, 100% relative humidity and a blot force of +20. Grids were vitrified by plunging into liquid ethane using a Vitrobot Mark IV device (Thermo Scientific).

Movies were recorded using a Titan Krios G3i microscope operated at 300 kV (Thermo Scientific) and equipped with a Gatan BioQuantum imaging filter using a 30 eV slit width and a K3 direct electron detector. Data were collected at a nominal magnification of 105,000×in electron counting mode using aberration-free image-shift (AFIS) correction in EPU (Thermo Scientific). Applied parameters are listed in *Table 1*.

After using CryoSPARC live (*Punjani et al., 2017*) for on-the-fly processing data to check data quality the full dataset was processed using RELION-3.1 (*Scheres, 2012*; *Zivanov et al., 2018*). Beam-induced motion was corrected using MOTIONCOR2 (*Zheng et al., 2017*) and dose-weighted images were generated from movies for initial image processing. Initial CTF parameters for each movie were estimated using Gctf algorithms (*Zhang, 2016*). Particles were picked with crYOLO (*Wagner et al., 2019*) using the neural network trained general model approach and cleaned *via* 2D classification in RELION-3.1. Unsupervised Initial model building, 3D classification, 3D refinement, along with post-processing steps like CTF refinement, Bayesian polishing and final map reconstructions were performed with RELION-3.1. Maps were visualized with Chimera (*Pettersen et al., 2004*) and models automatically built with Arp/Warp (*Langer et al., 2008*) or manually within COOT (*Emsley and Cowtan, 2004*). Further real-space refinement was performed using PHENIX (*Adams et al., 2010*).

## Software programs for structural analysis and presentation

The quality of the model was assessed with COOT and MolProbity (*Williams et al., 2018*). Interface area was calculated with PISA (*Krissinel and Henrick, 2007*) and coordinate superposition with COOT or DALI (*Holm, 2020*). For the AlphaFold2 calculations, the EtfA sequence and the EtfA and EtfB sequences stringed together were used as the only input (*Jumper et al., 2021*). *Figures 4C–9B* as well as *Figure 1—figure supplement 1*, *Figure 4—figure supplement 1* and *3* and *Figure 5—figure supplement 1* and *2*, Video 1 were produced with Chimera.

# Acknowledgements

Work in the laboratory of VM was supported by the Deutsche Forschungsgemeinschaft (DFG). KK thanks the International Max Planck Research School and Hartmut Michel for funding. We further thank Hartmut Michel, Janet Vonck and Werner Kühlbrandt for generous support.

## Additional information

### Funding

| Funder | Grant reference number | Author |
|---|---|---|
| International Max Planck Research School for Advanced Methods in Process and Systems Engineering | | Kanwal Kayastha |
| Deutsche Forschungsgemeinschaft | | Alexander Katsyv Volker Müller |

The funders had no role in study design, data collection and interpretation, or the decision to submit the work for publication.

### Author contributions

Kanwal Kayastha, Data curation, Formal analysis, Investigation, Methodology, Validation, Visualization, Writing – original draft, Writing – review and editing; Alexander Katsyv, Data curation, Investigation, Methodology, Supervision, Validation, Visualization, Writing – original draft, Writing – review and editing, Formal analysis; Christina Himmrich, Data curation, Investigation; Sonja Welsch, Methodology, Supervision; Jan M Schuller, Methodology; Ulrich Ermler, Conceptualization, Data curation, Funding acquisition, Investigation, Supervision, Visualization, Writing – original draft, Writing – review and editing; Volker Müller, Conceptualization, Funding acquisition, Project administration, Supervision, Validation, Writing – original draft, Writing – review and editing

### Author ORCIDs

Kanwal Kayastha (iD) http://orcid.org/0000-0001-7910-9743
Alexander Katsyv (iD) http://orcid.org/0000-0001-9212-231X
Ulrich Ermler (iD) http://orcid.org/0000-0002-9583-1418
Volker Müller (iD) http://orcid.org/0000-0001-7955-5508

### Decision letter and Author response

Decision letter https://doi.org/10.7554/eLife.77095.sa1
Author response https://doi.org/10.7554/eLife.77095.sa2

## Additional files

### Supplementary files

- Supplementary file 1. I. Table of corresponding primers used. II. Cloning of pET21a_lctBC-StrepD.
- Transparent reporting form
- Source data 1. SDS-PAGE, native PAGE and TLC plate or *Figure 2*.

### Data availability

The cryo-EM map and coordinates were deposited under accession numbers EMD-13960 and 7QH2, respectively.

The following datasets were generated:

| Author(s) | Year | Dataset title | Dataset URL | Database and Identifier |
|---|---|---|---|---|
| Kaystha K, Ermler U | 2022 | Structure of the Ldh-EtfAB complex | https://www.ebi.ac.uk/emdb/EMD-13960 | Electron microscopy data bank, EMD-13960 |
| Kaystha K, Ermler U | 2022 | Structure of the Ldh-EtfAB complex | https://www.rcsb.org/structure/7QH2 | RCSB Protein Data Bank, 7QH2 |

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
