## [Editor Report]

This paper describes a high-resolution cryoEM structure of the lactate dehydrogenase-electron transferring flavoprotein complex that provides critical insight on a basic metabolic pathway of anaerobic fermentation. What is novel is that this is the first structure of a complex wherein the dehydrogenase runs as such, rather than as a reductase. The lactate produced by glycolysis is oxidized to pyruvate with concomitant reduction of NAD^+^ to NADH. Because electrons are being supplied by the dehydrogenase, the ETF executes confurcation in contrast to all of those elucidated so far, which function in the opposite direction to effect bifurcation.

---

## [Decision Letter]

**Decision letter after peer review:**

Thank you for submitting your article "Structure-based electron-confurcation mechanism of the Ldh-EtfAB complex" for consideration by *eLife*. Your article has been reviewed by 3 peer reviewers, including Pimchai Chaiyen as the Reviewing Editor and Reviewer #1, and the evaluation has been overseen by Philip Cole as the Senior Editor. The following individual involved in the review of your submission has agreed to reveal their identity: Andrea Mattevi (Reviewer #3).

This paper describes a high-resolution cryoEM structure of the lactate dehydrogenase-electron transferring flavoprotein complex that provides critical insight into the basic metabolic pathway of anaerobic fermentation. What is novel is that this is the first structure of a complex wherein the dehydrogenase runs as such, rather than as a reductase. The lactate produced by glycolysis is oxidized to pyruvate with concomitant reduction of NAD^+^ to NADH. Because electrons are being supplied by the dehydrogenase, the ETF executes confurcation in contrast to all of those elucidated so far, which function in the opposite direction to effect bifurcation. As electron-confurcation and electron-bifurcation have emerged as important paradigms of cellular bioenergetics, the data reported herein pave a way for future exploration of similar electron transfer systems. The findings reported here can lay the ground for understanding structural biology related to these intriguing bioenergetic systems and may possibly be mentioned in a general textbook in the future.

General clarifications:

The manuscript needs considerable improvement with an eye to the readers who are not familiar with the electron bifurcation and other known electron bifurcating complexes.

The term shuttle is used to mean one domain of the ETF. Please provide a definition at the time of first use, so that this is clear. (It is well identified in Figure 4, but figure 4 is not mentioned for several pages after this terminology's first use.)

Specific comments:

1. It is interesting to see the impact that alphafold can have on experimental structure analysis. However, as a general comment, the authors tend to use the alphafold structures almost uncritically, as if they were experimentally determined. The manuscript should refer to the quality indicators given by alphafold and indicate how reliable the models are in particular around the regions that are mechanistically discussed in the text.

2. The last paragraph of the main text (lines 446-460) should be mostly re-written and clarified. The sentence "The midpoint redox potential of the pyruvate/lactate pair is -190 mV, which is substantially lower than that of the crotonyl-CoA/butyryl-CoA pair of 10 mV, which turns a bifurcation into a confurcation event" requires a better explanation as, here, we are at the heart of the functional and structural principles governing the LDH complex. Figure 7B remains thereby difficult to be fully understood. Likewise, the sentence "When assuming that the redox potentials of the electron carriers between b-FAD and lactate also differ by ca. 180 mV the microenvironment of a-FAD and d-FAD should be distinguished between Bcd-EtfAB and Ldh-EtfAB (Figure 7C)." remains honestly quite obscure to me. Likewise, the text should improve the discussion about the role of Gly37 and Gly139 and why the flavin of LDH should feature a lower redox potential compared to the analogous flavin of Bcd.

3. The manuscript should explain more clearly and from the beginning, the difference between confurcation and bifurcation. Likewise, the introduction should be accompanied by a scheme depicting the overall mechanism of electron bifurcation/confurcation in general and with reference to the reaction by the LDH system.

4. The abstract should be improved. Specifically, the text at lines 38-42 will be difficult to follow, at least for the casual reader.

5. Figure 2 legend, explain the meaning of CODH.

6. Line 189, the meaning of "promiscuous" EtfAB should be described explicitly.

7. Line 306, please describe where the residues 122-130 are located in the structure of the complex and why they define an electron-bifurcating system. Why are these residues making the difference between bifurcating and non-bifurcating enzymes?

8. The supplement figures of Table 1 should probably be moved to the main text as they are crucial for understanding the rationale of the mutagenesis experiments.

9. Line 322, for the analysis of the catalytic role of Arg205, the authors may refer to recently published work (doi.org/10.1016/j.jbc.2022.101733).

10. Line 180 and throughout, CarCDE and Fix-EtfABX should be described in more detail as the authors often refer to them for their structural and functional comparisons.

11. Line 437, can the authors comment on the proton sources that couple protonation and reduction of the flavins.

12. Figure 7, the legend states that the two-electron reduced FAD is shown in orange but the drawing shows it colored in red.

13. Figure 7 legend, what is an "escapement" mechanism? Explain.

14. The figure supplement 1 of Figure 4 does not show the residues listed in the figure legend (e.g. Leu195, Tyr297 etc).

15. Line 102, rephrase, for example replacing 'holded' by 'held'. (English is irrational.)

16. Line 125: Gel filtration is two words not one.

17. Figure 1C, how were the flavins visualized?

18. Line 144: Remove period after 'U/mg'.

19. Line 165: BS3 is used but not explained until the methods section. For all non-standard abbreviations please define abbreviation at time of first use. This is particularly important for this abbreviation, which appears to be the abbreviation BS with a reference to footnote number 3 (non-existent).

20. Line 182, may I suggest 'dehydrogenase coupled' or 'dehydrogenase connected' as conducting does not quite fit here.

21. Figure 3C. This figure is very attractive, but the lettering does not quite match the ribbons with respect to color. Please use the dropper tool to achieve a match. This request is made only because the figure is otherwise so attractively colored. I have not devoted the same attention to average figures. Also please say what software was used to generate the figures, especially if the software is provided for free (support the authors with generous credit).

22. Numbering of the supplemental figures and tables is very confusing. Please assemble these in a single document and use each numerical designation only once. E.g.: please only call one figure S1A. Please use the term 'figure' for all graphics and reserve the term 'table' for all numerical and otherwise tabulated information (e.g., 'Figure – Supplement 1B' is also Figure 3).

Line 281 advertises the existence of another figure, but no figure number is given, my guess is that Figure S5 is intended.

23. Line 303, please provide a literature reference for the 'single nucleotide exchange' method. Some of the amino acid substitutions described would appear to require that more than one nucleotide be changed, so I infer a clever new method.

24. Line 304 'supplementary File' please refer to a specific supplementary table or figure.

25. Line 320, the Δ205 notation is used, but I believe R205A is intended. Please reserve the Δ symbol for deletions. Similarly, on line 332, are these deletions? Substitutions are described in the supporting table.

26. Line 331, 'farer' should possibly be 'farther'?

27. Lines 340-341, Thank you for this. Please cite other examples in the literature.

28. Line 468: 'red and orange' should be 'orange and red' to agree with the oxidation states listed earlier in the line.

29. Lines 667, 671. Duplicates of the same reference.

30. Line 743, please provide complete reference information

31. Lines 776 and 782. Duplicates of the same reference.

32. Lines 621, 622, 643, 795, flawed carriage return.

33. Page 34: why is the first figure in the supplement Figure 3? Please call it figure S1 (supplemental figure 1). Why is a table also called Figure 3?

34. Page 42: Table A (primers). What distinguishes the lower-case nucleotides from those listed in upper-case? I could not see that the reverse primers were the reverse complements of the forward primers. Please explain.

35. Page 5, why does purification need to be done anaerobically? Was the rest of structure determination handled anaerobically as well?

36. In Figure 1B, why does the native PAGE of (Ldh-EtfAB)2 show the bands around 220 and 280 kDa instead of 253 kDa (Line 121). What are these bands? The authors should improve the quality of bands shown in the native PAGE by performing the analysis again.

37. As the complex can easily dissociate, has the Kd value been estimated?

38. Why did the authors not try to determine individual structures of EtfAB? By adding NAD+, would it be possible to obtain the X-ray determined structure in addition to ALphaFold calculations?

39. Line 147, typo "its highest.."

40. Line 148, I am curious about the data showing that the FBEC activity was inactive at pH 5. In real situations, what is the cellular pH of A. woodii? I think such microbes would have pH around that range. If FBEC is inactive at pH5, how can it conduct its reactions inside the cell?

41. Figure 2, explanation regarding pH-dependent activity profiles should also be given.

42. Were the assays done under the optimized conditions giving the highest activities?

43. Page 8, how can one be sure that the cross-linking by BS does not create artifacts?

44. Page 13-14, a diagram explaining assays in Table 2 should be helpful for the reader to understand.

45. Page 14, "..variant consisting of the D122A, D124A, T125G, Q127G, V128A and 311 P130A substitutions shows a lower overall FAD amount of 1.7……….oxidoreductase activities were reduced to 7% and not anymore measurable, respectively. Likewise, the NADH:DCPIP oxidoreductase activity, measuring the hydride transfer from NADH to b-FAD, was also abolished. Both findings can be explained by the 316 expected strong decrease of the b-FAD content……"

Was the decrease in activities mentioned proportional to the amount of FAD lost? One can calculate whether activities detected are according to the % FAD retained?

46. Page 14-15, for the abolishment of the hydride transfer activity in the NADH:DCIP assays, have the authors tried to measure direct hydride transfer to b-FAD. Can this be done by monitoring FAD reduction under anaerobic conditions?

47. Line 367, the iron in [ΔFe/S-domain] and [ΔFe/S] variants are completely lost, but why does the overall yield increase 4-fold.

48. Line 442, regarding the statement that external Fdred binding may induce conformational changes to reduce their distance to ca. 14 Å, any evidence to support this?

49. Line 464, regarding the statement that Glu37 and Glu139 are only found in the Ldh-EtfAB complex and these residues may destabilize the electron-rich reduced state of flavin, it would be better to have a reference to endorse this statement.

[Editors' note: further revisions were suggested prior to acceptance, as described below.]

Thank you for resubmitting your work entitled "Structure-based electron-confurcation mechanism of the Ldh-EtfAB complex" for further consideration by *eLife*. Your revised article has been evaluated by Philip Cole (Senior Editor) , a Reviewing Editor, and an external reviewer.

The manuscript has been improved but there are some remaining issues that need to be addressed, as outlined below:

*Reviewer #1 (Recommendations for the authors):*

The manuscript has been improved and read better than the previous version.

Perhaps, the authors may consider the following suggestions to make them clearer.

1. I find Figure 1 confusing and too crowded. The authors may consider drawing Two separate panels (left and right) to explain different scenarios of confurcation and bifurcation. Do not mix them in the same figure.

2. The authors may consider adding more responses to reviewers as a brief explanation of the manuscript as well. This will be clearer to readers.

*Reviewer #3 (Recommendations for the authors):*

The authors have carefully revised the manuscript according to the reviewers' suggestions and advice. The reported results are of interest and add to our understanding of electron bifurcation.

The authors may notice that the sentences on lines 174-177, 182-185, and 269 should be better re-written for the sake of clarity.

---

## [Author Response]

General clarifications:The manuscript needs considerable improvement with an eye to the readers who are not familiar with the electron bifurcation and other known electron bifurcating complexes.The term shuttle is used to mean one domain of the ETF. Please provide a definition at the time of first use, so that this is clear. (It is well identified in Figure 4, but figure 4 is not mentioned for several pages after this terminology's first use.)Specific comments:1. It is interesting to see the impact that alphafold can have on experimental structure analysis. However, as a general comment, the authors tend to use the alphafold structures almost uncritically, as if they were experimentally determined. The manuscript should refer to the quality indicators given by alphafold and indicate how reliable the models are in particular around the regions that are mechanistically discussed in the text.

We have used as input only the sequence of EtfA and did not work with any template. As result we took the pdb file with the highest quality values (new Figure 5 – supplement 2). The quality plot suggests a reliable prediction throughout the chain (except for the last 30 residues), that also includes the segment forming the interface. The rms deviation between the Alphafold EtfA model and the experimental model was 1.31 Å for the EtfA base (domain I) and 0.98 Å for EtfA shuttle domain (domain II) indicating a high agreement. A different angle between the EtfAB base and the EtfAB shuttle domain was expected in comparison to that of the experimentally analyzed D-state as EtfB and Ldh was not included in the AlphaFold calculation. The new interface between the EtfAB base and the EtfAB shuttle domain is plausible. Several hydrophobic and hydrophilic interactions are formed shown in the new Figure 5 – Supplement 1C. The buried area is ca. 700 Å2.

2. The last paragraph of the main text (lines 446-460) should be mostly re-written and clarified. The sentence "The midpoint redox potential of the pyruvate/lactate pair is -190 mV, which is substantially lower than that of the crotonyl-CoA/butyryl-CoA pair of 10 mV, which turns a bifurcation into a confurcation event" requires a better explanation as, here, we are at the heart of the functional and structural principles governing the LDH complex. Figure 7B remains thereby difficult to be fully understood. Likewise, the sentence "When assuming that the redox potentials of the electron carriers between b-FAD and lactate also differ by ca. 180 mV the microenvironment of a-FAD and d-FAD should be distinguished between Bcd-EtfAB and Ldh-EtfAB (Figure 7C)." remains honestly quite obscure to me. Likewise, the text should improve the discussion about the role of Gly37 and Gly139 and why the flavin of LDH should feature a lower redox potential compared to the analogous flavin of Bcd.

We have completely rewritten the final paragraph of the manuscript.

3. The manuscript should explain more clearly and from the beginning, the difference between confurcation and bifurcation. Likewise, the introduction should be accompanied by a scheme depicting the overall mechanism of electron bifurcation/confurcation in general and with reference to the reaction by the LDH system.

We have added a new figure 1 into the introduction part involving also the Ldh-EtfAB complex as suggested.

4. The abstract should be improved. Specifically, the text at lines 38-42 will be difficult to follow, at least for the casual reader.

We modified lines 38-42 in the abstract but there is not much space for explaining.

5. Figure 2 legend, explain the meaning of CODH.

Carbon monoxide dehydrogenase (CODH) was used to keep the amount of reduced ferredoxin in the FBEC assay constant. The term CODH is now explained in the figure legend. The importance of CODH in the assay is explained in the material and methods part. (see also item 44)

6. Line 189, the meaning of "promiscuous" EtfAB should be described explicitly.

“Promiscuous” means in this context that several dehydrogenase types can associate with the same EtfAB complex. In the manuscript we have replaced the word “promiscuous”.

7. Line 306, please describe where the residues 122-130 are located in the structure of the complex and why they define an electron-bifurcating system. Why are these residues making the difference between bifurcating and non-bifurcating enzymes?

In non-bifurcating EtfAB the b-FAD is replaced by an AMP. Residues 122-130 flank the isoalloxazine ring in bifurcating EtfAB, which is absent in non-bifurcating EtfAB. We showed the position of residues 122-130 in Figure 7. The residues relevant for FBEB are described in the review of Buckel and Thauer, 2018. This review is cited at this position.

8. The supplement figures of Table 1 should probably be moved to the main text as they are crucial for understanding the rationale of the mutagenesis experiments.

We moved this figure into the main text after Table 1 as figure 7.

9. Line 322, for the analysis of the catalytic role of Arg205, the authors may refer to recently published work (doi.org/10.1016/j.jbc.2022.101733).

We integrated this reference into the text.

10. Line 180 and throughout, CarCDE and Fix-EtfABX should be described in more detail as the authors often refer to them for their structural and functional comparisons.

We made a supplementary figure showing the structures of Bcd-EtfAB, CarCDE and Fix-EtfABC and Ldh-EtfAB (Figure 1 —figure supplement 1). One EtfAB unit is in the same orientation in all structures.

11. Line 437, can the authors comment on the proton sources that couple protonation and reduction of the flavins.

There is no obvious adjacent amino acid suitable for accepting/donating a proton from/to N5 of b-FAD. There is, of course, Arg205, which is not a suitable proton donor but it is in direct contact to N5 and the bulk solvent and may act as mediator. It is unclear yet whether NADH binding completely shields N5. There might be an access from the NAD binding site.

12. Figure 7, the legend states that the two-electron reduced FAD is shown in orange but the drawing shows it colored in red.

Thank you, we have corrected.

13. Figure 7 legend, what is an "escapement" mechanism? Explain.

We explained the escapement mechanism in the legend of Figure 1B.

14. The figure supplement 1 of Figure 4 does not show the residues listed in the figure legend (e.g. Leu195, Tyr297 etc).

In this figure the overall view of the interface is more important than individual residues. We changed the figure in a way to take into accounts the comment and to keep the figure clear.

15. Line 102, rephrase, for example replacing 'holded' by 'held'. (English is irrational.)

Corrected.

16. Line 125: Gel filtration is two words not one.

Corrected.

17. Figure 1C, how were the flavins visualized?

Flavins were visualized under UV light. We added a sentence in the figure legends.

18. Line 144: Remove period after 'U/mg'.

Corrected.

19. Line 165: BS3 is used but not explained until the methods section. For all non-standard abbreviations please define abbreviation at time of first use. This is particularly important for this abbreviation, which appears to be the abbreviation BS with a reference to footnote number 3 (non-existent).

The cryo-EM structure of the fragile Ldh-EtfAB complex from *A. woodii* was determined from a sample cross-linked with bis (sulfosuccinimidyl) suberate (BS^3^) which is one of the most used agents for this purpose. The upper 3 belongs to the name of the cross-linker.

20. Line 182, may I suggest 'dehydrogenase coupled' or 'dehydrogenase connected' as conducting does not quite fit here.

We changed to ‘dehydrogenation-connected’. We wish to say that the shuttle a-FAD is in productive electron transfer distance to the d-FAD of the dehydrogenase.

21. Figure 3C. This figure is very attractive, but the lettering does not quite match the ribbons with respect to color. Please use the dropper tool to achieve a match. This request is made only because the figure is otherwise so attractively colored. I have not devoted the same attention to average figures. Also please say what software was used to generate the figures, especially if the software is provided for free (support the authors with generous credit).

We improved this figure, adapted the letter and the colors. We mentioned in line 616 that Chimera was used to produce this figure.

22. Numbering of the supplemental figures and tables is very confusing. Please assemble these in a single document and use each numerical designation only once. E.g.: please only call one figure S1A. Please use the term 'figure' for all graphics and reserve the term 'table' for all numerical and otherwise tabulated information (e.g., 'Figure – Supplement 1B' is also Figure 3).Line 281 advertises the existence of another figure, but no figure number is given, my guess is that Figure S5 is intended.

We collected all Figure supplements and one supplementary figure in one file and tried to follow the *eLife* rules.

23. Line 303, please provide a literature reference for the 'single nucleotide exchange' method. Some of the amino acid substitutions described would appear to require that more than one nucleotide be changed, so I infer a clever new method.

This kind of amino acid substitutions were previously described in Demmer et al. 2018 (doi: 10.1002/1873-3468.12971). We added the reference.

24. Line 304 'supplementary File' please refer to a specific supplementary table or figure.

Supplementary file 1 stands alone and does not really fit to a specific figure or table. We saw in other *eLife* manuscripts a related handling of the Supplementary file. If this is not possible we will find a solution.

25. Line 320, the Δ205 notation is used, but I believe R205A is intended. Please reserve the Δ symbol for deletions. Similarly, on line 332, are these deletions? Substitutions are described in the supporting table.

We changed ∆205 and ∆189 to R205A and D189A.

26. Line 331, 'farer' should possibly be 'farther'?

Corrected.

27. Lines 340-341, Thank you for this. Please cite other examples in the literature.

We cited the structural papers of the EtfB family.

28. Line 468: 'red and orange' should be 'orange and red' to agree with the oxidation states listed earlier in the line.

Corrected.

29. Lines 667, 671. Duplicates of the same reference.

These are two different references from the same authors.

Corrected.

30. Line 743, please provide complete reference information

Müller, V. (2008) Bacterial Fermentation. Encyclopedia of Life Sciences. Chichester:

John Wiley and Sons, United Kingdom

31. Lines 776 and 782. Duplicates of the same reference.

Corrected.

32. Lines 621, 622, 643, 795, flawed carriage return.

Corrected.

33. Page 34: why is the first figure in the supplement Figure 3? Please call it figure S1 (supplemental figure 1). Why is a table also called Figure 3?

Because the table belongs to Figure 3. We renamed several figure supplements.

34. Page 42: Table A (primers). What distinguishes the lower-case nucleotides from those listed in upper-case? I could not see that the reverse primers were the reverse complements of the forward primers. Please explain.

The applied site-directed mutagenesis method does not require a reverse complement for ligation because it’s a blunt-end ligation. Therefore, the primers have also no reverse complement regions. Back-to-back primer design methods have the advantage of transforming non-nicked plasmids. In addition, because the primers do not overlap each other, deletions sizes are only limited by the plasmid and insertions are only limited by the constraints of modern primer synthesis. In the first step we amplified our genes using primers described in Table 1 – Supplement 1. The second step involves incubation with an enzyme mix containing a kinase, a ligase and DpnI. Together, these enzymes allow for rapid blunt-end circularization of the PCR product and removal of the template DNA. The last step was the transformation into chemically competent cells.

35. Page 5, why does purification need to be done anaerobically? Was the rest of structure determination handled anaerobically as well?

Expression and purification have to be done under anaerobic conditions because the [4Fe-4S] cluster is highly oxygen-sensitive. The grid freezing procedure, finished in ca. 1 min, was performed on air by using O_2_ free buffers. Kinetic FBEB or FBEC experiments have also to be done under anaerobic conditions as O_2_ competes with ferredoxin.

36. In Figure 1B, why does the native PAGE of (Ldh-EtfAB)2 show the bands around 220 and 280 kDa instead of 253 kDa (Line 121). What are these bands? The authors should improve the quality of bands shown in the native PAGE by performing the analysis again.

First, we have to apologize using a picture of such a bad quality gel. We repeat the native PAGE experiments in the revised manuscript. The clear bands of the new gel allowed us to recalculate the sizes of the Ldh/EtfAB complex in the native page, which are approximately ≈150 kDa and ≈265 kDa. It corresponds to the sizes of a monomeric and dimeric form. Due to its fragility the Ldh-EtfAB complex is partially dissociated in the native PAGE. In addition, gel filtration profiles clearly indicate the partial dissociation of the Ldh-EtfAB complex into Ldh and EtfAB. The migration of the Ldh-EtfAB complex as two different fragments with different sizes was already shown in the first publication of the purified complex (Weghoff et al. 2014; https://doi.org/10.1111/1462-2920.12493) and is obviously independent of the expression and purification procedure. We added Figure 1 – supplement 1 to show the disassembly.

37. As the complex can easily dissociate, has the Kd value been estimated?

No, this number would be only relevant for the *A. woodii* enzyme and does not bring us forward in understanding FBEB/FBEC.

38. Why did the authors not try to determine individual structures of EtfAB? By adding NAD+, would it be possible to obtain the X-ray determined structure in addition to ALphaFold calculations?

Structures of non-bifurcating and bifurcating EtfAB are already available in a physiologically incorrect B-like conformation. Previously, we did structure determination of *Acidaminococcus fermentans* EtfAB not only in the substrate-free form but also by soaking NAD^+^/NADH. But we always obtained the same non-physiological conformation. Moreover, we worked several years on the Bcd-EtfAB complex of *Clostridium difficile* to characterize the B state by X-ray and EM structural analysis but without success. Even when we had determined an EtfAB structure you could still be sceptic as the two flexible linkers might still not adjust the correct orientation between the EtfAB base and the EtfAB shuttle domain due to crystal lattice effects. Because of the non-success we thought that the correct B-conformation is only short-living and when attainable at all, then in the complete (Ldh-EtfAB)_2_ complex. Therefore, we focused on the (Ldh-EtfAB)_2_ complex in this work. The plausible B-state obtained with AlphaFold2 later on was a big surprise for us.

39. Line 147, typo "its highest.."

Corrected.

40. Line 148, I am curious about the data showing that the FBEC activity was inactive at pH 5. In real situations, what is the cellular pH of A. woodii? I think such microbes would have pH around that range. If FBEC is inactive at pH5, how can it conduct its reactions inside the cell?

The cellular pH of *A. woodii* has been not determined. However, the cellular pH of anaerobes is indeed not kept constant at around pH 7.0. Studies of cellular pH values in closely related species of *A. woodii*, were performed by Menzel and Gottschalk (1985). The extracellular pH of *Acetobacterium wieringae* decreased from 7.0 to 5.0 during growth on sugars; accordingly, the intracellular pH changed from 7.1 to 5.5 (∆pH between 0.1 and 0.5 (interior more alkaline)).

We determined the extracellular pH change of *A. woodii*. during growth on lactate. The extracellular pH of *A. woodii* decreased from 7.2 to 6.5 during growth on 80 mM lactate. According to *A. wieringae*, the internal pH of *A. woodii* growing on lactate should be around 7.3 to 7.0, which is optimal for the Ldh-Etf complex.

41. Figure 2, explanation regarding pH-dependent activity profiles should also be given.

The FBEC activity has a sharp maximum around pH 7.0. Due to the complexity of the system this result cannot directly be interpreted in structural terms. It is certainly compatible with the pK_a_ value of His423, the potential catalytic base for proton abstraction from lactate. Its pK_a_ might be higher than 6.0 toward the uncharged form in the protein interior.

42. Were the assays done under the optimized conditions giving the highest activities?

Yes, all activity assays were performed under optimized pH and temperature conditions.

43. Page 8, how can one be sure that the cross-linking by BS does not create artifacts?

We cannot exclude that the crosslinking experiment creates artefacts. BS^3^ is frequently used (https://doi.org/10.1016/j.cbpa.2020.07.008) and, so far, realistic results were obtained (e.g. https://doi.org/10.1101/2021.07.22.453353; https://doi.org/10.1016/j.cell.2020.02.034). The Ldh-EtfAB revealed a highly plausible architecture of the D-state and there is no doubt on its relevance. We would, however, not exclude that the cross-linker preferably stabilizes the shuttle domain in the resting D-state and therefore prevents populations of the B-state. An FBEC activity test resulted in 2.1 ± 0.1 U/mg corresponding to 20% turnover. Thus, the structure and function of the Ldh-EtfAB complex is not severely changed by cross-linking, in particular, when considering that the additional cross-linking and purification steps are stress situations for the fragile protein complex.

44. Page 13-14, a diagram explaining assays in Table 2 should be helpful for the reader to understand.

We have explicitly written in Author response image 1 the reactions of the assays, which can be, in principle, integrated into the manuscript. But we prefer to omit these reactions as they can be easily researched in the literature.

**Author response image 1. sa2fig1:** 

45. Page 14, "..variant consisting of the D122A, D124A, T125G, Q127G, V128A and 311 P130A substitutions shows a lower overall FAD amount of 1.7……….oxidoreductase activities were reduced to 7% and not anymore measurable, respectively. Likewise, the NADH:DCPIP oxidoreductase activity, measuring the hydride transfer from NADH to b-FAD, was also abolished. Both findings can be explained by the 316 expected strong decrease of the b-FAD content……"Was the decrease in activities mentioned proportional to the amount of FAD lost? One can calculate whether activities detected are according to the % FAD retained?

The methods applied for measuring the FAD content is not sufficiently sensitive to quantify small b-FAD amounts. It has to be considered that 3 FADs are involved and errors of ca. 10% are possible. The FBEC activity of 7% suggests a b-FAD content < 10% which is within the error of the FAD determination. No serious statement about the proportionality between activities and the amount of FAD is possible.

46. Page 14-15, for the abolishment of the hydride transfer activity in the NADH:DCIP assays, have the authors tried to measure direct hydride transfer to b-FAD. Can this be done by monitoring FAD reduction under anaerobic conditions?

In principle, it is possible to monitor b-FAD reduction with NADH alone. However, we have not done this experiment for Ldh-EtfAB. This type of experiment has been performed for EtfAB of *A.fermentans* (Chowdhury, N. P.,2014, doi.org/10.1074/jbc.M113.521013; Sucharitakul, J. 2020 doi:10.1111/febs.15343). We expect similar results for EtfAB of the Ldh-EtfAB complex.

47. Line 367, the iron in [ΔFe/S-domain] and [ΔFe/S] variants are completely lost, but why does the overall yield increase 4-fold.

We were also surprised to see an increase in the purification yield of [ΔFe/S-domain] and [ΔFe/S] variants. The same finding was made by the CarCDE complex (Demmer et al., 2018). We can only speculate that the production of Ldh-EtfAB becomes simplified when the [FeS] cluster has not to be formed. Its extraordinary flexibility might further delay the maturation process.

48. Line 442, regarding the statement that external Fdred binding may induce conformational changes to reduce their distance to ca. 14 Å, any evidence to support this?

No, we have no experimental evidence that supports this proposal. It is difficult to structurally characterize a FBEB enzyme in complex with an external ferredoxin as its binding is normally rather weak. It would be lucky to find a suitable model system. However, we have executed an Alphafold2 calculation with EtfA and EtfB (previously only with EtfA) and the Fd domain moved to EtfB. The distance between the [4Fe-4S] cluster and b-FAD is thereby shortened to ca. 14 Å. This result is not surprising when considering the high mobility of the Fd domain. Moreover, a related phenomenon was already observed twice in FBEB enzymes (DOI: 10.1002/1873-3468.12971; DOI: 10.1126/science.aan0425; DOI: 10.1126/science.abg5550). It appears to be useful for the FBEB/FBEC reaction that the electron transfer pathway between the external ferredoxin and the bifurcating flavin is only formed for the short period of the reaction. The interruption might also be a tool to ensure that only desired ferredoxins can properly dock to the surface and induce the conformational changes required for electron transfer.

49. Line 464, regarding the statement that Glu37 and Glu139 are only found in the Ldh-EtfAB complex and these residues may destabilize the electron-rich reduced state of flavin, it would be better to have a reference to endorse this statement.

For example, the following reference are relevant (Balland et al. 2009, doi.org/10.1021/ja806540j; Talfournier et al., 2001, doi.org/10.1074/jbc.M010853200 (elimination of a positive charge); Mothersole et al. 2019, doi: 10.1002/pro.3661).

There are mutagenesis data of charged residues adjacent to flavins but they will neither validate nor falsify our conclusion. These mutagenesis experiments are extremely delicate as small secondary changes at the isoalloxazine ring or its surrounding overcompensate the effect of pure mutation. A strong argument for the importance for Glu37 and Glu139 is their strict conservation in bifurcating Ldh but not in membrane-spanning Ldhs. Two acidic amino acids flanking two isoalloxazine group is a very unusual feature in biochemistry.

[Editors' note: further revisions were suggested prior to acceptance, as described below.]

The manuscript has been improved but there are some remaining issues that need to be addressed, as outlined below:Reviewer #1 (Recommendations for the authors):The manuscript has been improved and read better than the previous version.Perhaps, the authors may consider the following suggestions to make them clearer.1. I find Figure 1 confusing and too crowded. The authors may consider drawing Two separate panels (left and right) to explain different scenarios of confurcation and bifurcation. Do not mix them in the same figure.

Amended legend for Figure 1:

“Figure 1. Thermodynamic basis. (A) The FBEB reaction (1). It involves two reduction reactions (2 + 3) using the same electron donor (M) of medium redox potential and two electron acceptors (N and Fd) of high and low redox potential. Fd can be only replaced by flavodoxin (Chowdhury et al., 2016). Biochemical reactions are normally characterized by a pair-wise exchange of electrons between substrates that are protonated in their reduced state. FBEB occur when the positive difference redox potential ∆E_2_ between the strong electron acceptor (N) and the medium electron donor (MH) has a higher absolute value than the negative ∆E_3_ between Fd_ox_ and MH. In other words, ∆E_1_ has to be positive resulting in a negative Gibbs free energy (∆G = -nF∆E, n: moles of electrons exchanged, F: Faraday constant). As example, the electrochemical treatment and the thermodynamic scheme was provided for the reaction of the bifurcating Bcd-EtfAB complex, in which the b-FAD reduced by NADH endergonically donates one electron *via* a-FAD to d-FAD and then exergonically one electron to Fd_ox_. (B) The reverse FBEC reaction (1) calalyzing two oxidation reaction (2+3) with the same electron acceptor (M). Here, positive ∆E_3_ has a higher absolute value than the negative ∆E_2._ In the Ldh-EtfAB reaction low-potential Fd_red_ donates an electron to b-FAD *via* one [4Fe-4S] cluster in an endergonic reaction that is driven by the exergonic ET from d-FAD, *via* a-FAD to b-FAD•^−^. The generated b-FADH^‒^ transfers a hydride to NAD^+^. For the FBEB/FBEC process the first uphill ET step to b-FAD is reversed except when instanteneously pulled out by the second downhill ET step termed as escapement-type mechanism (Baymann et al., 2018). This tight coupling implicates catalytic inactivity in the absence of one of the three substrates and prevention of short circuit reactions e.g. from b-FAD•^−^ to a-FAD•^−^ and undesirable side reaction of the highly reactive b-FAD•^−^”.

2. The authors may consider adding more responses to reviewers as a brief explanation of the manuscript as well. This will be clearer to readers.Reviewer #3 (Recommendations for the authors):The authors have carefully revised the manuscript according to the reviewers' suggestions and advice. The reported results are of interest and add to our understanding of electron bifurcation.The authors may notice that the sentences on lines 174-177, 182-185, and 269 should be better re-written for the sake of clarity.

Line 174-179 (now 172-174)

“In contrast, non-bifurcating oxidoreductase-EtfAB complexes form a transient D-like state, which is beneficial as the EtfAB shuttle domain assembles with different dehydrogenase partners (Leys et al., 2003).”

Line 182-185 (now 178-180)

“The EtfAB shuttle domain swings ca. 75° from the D-state into a B-state position (*Figure 5 – video 1*) that deviates from the previously proposed one (Demmer et al., 2017).”

Line 269 (now 264-266)

“D189 of EtfA is 11 Å apart from b-FAD (*Figure 7*) and in the B-state 4.5 and 10 Å apart from the previously proposed and AlphaFold2-calculated a-FAD position, respectively (Demmer et al., 2017; Jumper et al., 2021).”